# DiffAdvMAP: Flexible Diffusion-Based Framework for Generating Natural Unrestricted Adversarial Examples

Zhengzhao Pan [1]    Hua Chen [2]    Xiaogang Zhang [1]

## Abstract

Unrestricted adversarial examples(UAEs) have posed greater threats to deep neural networks(DNNs) than perturbation-based adversarial examples(AEs) because they can make extensive changes to images without being restricted in a fixed norm perturbation budget. Although current diffusion-based methods can generate more natural UAEs than other unrestricted attack methods, the overall effectiveness of such methods is restricted since they are designed for specific attack conditions. Additionally, the naturalness of UAEs still has room for improvement, as these methods primarily focus on leveraging diffusion models as strong priors to enhance the generation process. This paper proposes a flexible framework named Diffusion-based Adversarial Maximum a Posterior(DiffAdvMAP) to generate more natural UAEs for various scenarios. DiffAdvMAP approaches the generation of UAEs by sampling images from posterior distributions, which is achieved by approximating the posterior distribution of UAEs using the prior distribution of real data learned by the diffusion model. This process enhances the naturalness of the UAEs. By incorporating an adversarial constraint to ensure the effectiveness of the attack, DiffAdvMAP exhibits excellent attack ability and defense robustness. A reconstruction constraint is designed to enhance its flexibility, which allows DiffAdvMAP to be tailored to various attack scenarios. Experimental results on ImageNet show that we achieve a better trade-off between image quality, flexibility, and transferability than baseline unrestricted adversarial attack methods.

## 1. Introduction

Deep Neural Networks (DNNs) have been prosperous in various vision tasks these years, such as object detection, face recognition, and semantic segmentation. However, many works have shown that DNNs are vulnerable to adversarial examples. Adversarial examples are images intentionally crafted by adding tiny perturbations to natural images. Such modified images can deceive DNNs to make wrong predictions while remaining imperceptible to humans, bringing security risks to decision-critical systems. This vulnerability poses great threats to lots of vision tasks such as image classification (Goodfellow et al., 2014), (Carlini & Wagner, 2017), (Madry et al., 2017), segmentation (Li et al., 2023), and tracking (Li et al., 2021b), (Li et al., 2023).

Unlike traditional perturbation-based adversarial examples (AEs), which limit perturbations to a small range to maintain imperceptibility, unrestricted adversarial examples (UAEs) are generated by applying extensive natural transformations to images, such as color conversion, which significantly reduces the noticeable noise patterns in AEs. UAEs can also be generated by training a generative model like AC-GAN (Song et al., 2018), enabling the attacker to produce a more natural and unlimited number of UAEs. Such approaches do not require perturbing real images with restricted perturbations, thus being more concealed and effective than traditional AEs. As a result, UAEs have emerged as a significant area of study in adversarial examples over the past few years due to their potential threat to deep neural networks. Though generative models like GANs and VAEs can learn and sample from data distribution effectively, it's difficult to perform well on complex and high-quality datasets like ImageNet (Deng et al., 2009) because of their weak interpretability. Diffusion models (Ho et al., 2020) have shown their superiority in synthesizing realistic and high-quality images in recent years, thus, they become powerful competitors to GANs and VAEs in generating UAEs. Inspired by this, recent works (Dai et al., 2025), (Chen et al., 2023), (Liu et al., 2023a) explore new methods to generate realistic UAEs on complex datasets with diffusion models and obtain better performance than previous works.

However, there still exist some problems that affect the effectiveness and naturalness of UAEs generated by diffu-

---

[1]College of Electrical and Information Engineering, Hunan University, Changsha, China [2]College of Computer Science and Electronic Engineering, Hunan University, Changsha, China. Correspondence to: Hua Chen <chua@hnu.edu.cn>, Xiaogang Zhang <zhangxg@hnu.edu.cn>.

*Proceedings of the 42$^{nd}$ International Conference on Machine Learning*, Vancouver, Canada. PMLR 267, 2025. Copyright 2025 by the author(s).

sion models to be considered: 1) As is shown in (Meng et al., 2021), diffusion models tend to add low-level semantic information such as the layout in the early generation steps while more high-level semantic information in the later steps, modifying the latent code of the early steps in the generation process(Chen et al., 2023) may change the low-level features and take the risk of generating unnatural UAEs. Although some methods(Dai et al., 2025)(Chen et al., 2024a) generate UAEs by generating adversarial high-level features, they primarily focus on using diffusion models as strong priors to enhance the generation process. This approach does not fully leverage the prior knowledge of real data distributions learned by diffusion models, which may still take the risk of generating unrealistic features. 2) Most existing methods are limited to a fixed set of scenarios, as they are designed for specific attacking conditions, such as generating UAEs similar to given reference images or producing UAEs from noise. This narrow focus restricts the overall effectiveness of the adversarial examples.

To this end, we propose a flexible diffusion-based unrestricted adversarial attack framework to generate natural UAEs. In our opinion, the posterior distribution of UAEs derived from the prior distribution of natural data learned by the diffusion model is more close to natural data distribution, we can generate more natural UAEs by sampling from this distribution. We leverage the generation process of a pre-trained diffusion model, extending and adjusting the maximum a posterior(MAP) method to form our DiffAdvMAP framework to generate natural UAEs. Under the Bayesian framework, we first derive the posterior distribution of UAEs based on the real data distribution learned by the diffusion model under the adversarial and reconstruction constraints, the adversarial constraint is used to ensure the effectiveness of the attack, and the reconstruction constraint is used to control the content of generated UAEs. Then we go through the generation process of the diffusion model and sample UAEs from such distribution. Since our framework samples UAEs from the approximated posterior distribution of UAEs, there's no need to go through the whole generation process to remove too many conspicuous adversarial noises. We integrate a destruction and construction method into our framework, which destroys most high-level features of real images by the diffusion process, and regenerates adversarial features via DiffAdvMAP. As a result, our framework can generate UAEs with a truncated generation process while protecting most low-level features, thus improving the naturalness and generation speed of UAEs. Finally, when facing different attacking tasks such as generating UAEs similar to the given images, generating UAEs from noise, generating UAEs via regenerating specified regions of given images, and generating UAEs via changing the color or style of given images, the reconstruction constraint in DiffAdvMAP can be customized to such tasks while keeping the naturalness.

Our main contributions are summarized as follows:

- We propose a flexible diffusion-based framework for generating UAEs named DiffAdvMAP, it can generate UAEs under various attacking conditions. We achieve it by approximating the posterior distribution of UAEs using pre-trained diffusion models and sampling from the distribution. This approach leads to a better naturalness than most diffusion-based attack methods.

- We design an adversarial constraint and a reconstruction constraint within the Bayesian framework to generate UAEs. The adversarial constraint ensures the effectiveness of UAEs; the reconstruction constraint grants our framework the flexibility to handle various attack conditions.

- Experimental results regarding white-box attack success rate, transferability, and defense robustness demonstrate the effectiveness of DiffAdvMAP. Additionally, UAEs generated under various attack conditions further emphasize its superiority in flexibility and effectiveness over baseline attacks.

## 2. Related Works

### 2.1. Adversarial Examples

Perturbation-based adversarial attacks are performed by adding small and imperceptible perturbations to natural images such that the target model makes wrong predictions. Since (Szegedy et al., 2013) shows the existence of adversarial examples, the security concerns of such attacks are increasing in computer vision and machine learning communities as more and more advanced and powerful methods are developed (Moosavi-Dezfooli et al., 2016)(Long et al., 2022). On the other hand, adversarial attacks play important roles in improving contrastive learning(Lee et al., 2020)(Ho & Nvasconcelos, 2020), image recognition(Xie et al., 2020), privacy protection(Li et al., 2021a)(Liu et al., 2023a), and other applications. Attackers can easily generate perturbation-based adversarial samples by using gradient-based methods such as fast gradient sign method(FGSM)(Goodfellow et al., 2014), CW attack(Carlini & Wagner, 2017), projected gradient descent(PGD)(Madry et al., 2017).

While most of the perturbation-based adversarial attacks that focus on optimizing additive perturbations at the pixel level have achieved good results, it is shown that the restrictions of perturbations are not accurate in representing the way that humans perceive the differences between similar images (Jia et al., 2022), (Yuan et al., 2022), and thus introducing conspicuously noise patterns, such as the global noise introduced by the PGD attack. As a result, researchers

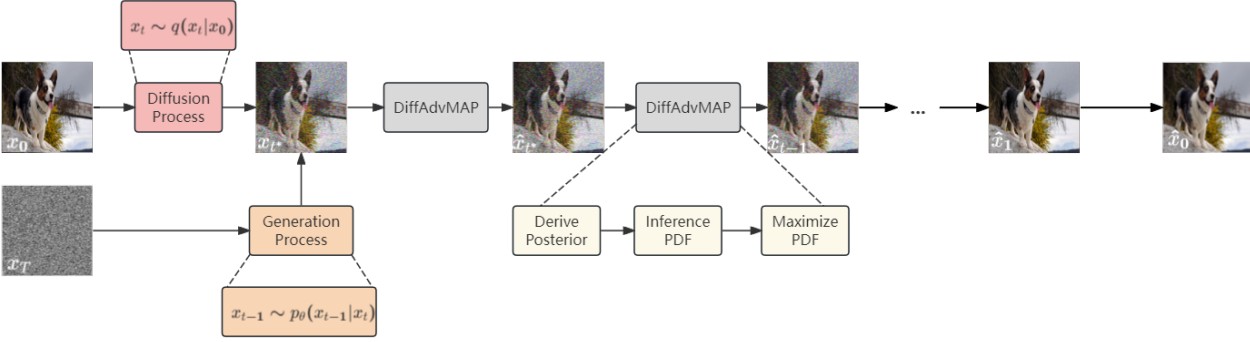

*Figure 1.* An overview of the DiffAdvMAP algorithm for generating unrestricted adversarial examples

turn to generating AEs using generative models, achieving higher realism than perturbation-based AEs while keeping a high success rate. (Wong & Kolter, 2020) trains a conditional VAE to generate a variety of perturbations. (Xiao et al., 2018) trains a conditional GAN to produce adversarial examples directly. (Song et al., 2018) trains an AC-GAN and samples adversarial examples from noise. (Qiu et al., 2020) generates imperceptible AEs by modifying the attributes of the natural images using a GAN. (Bhattad et al., 2019) perturbs images from the perspective of color and texture by leveraging corresponding pre-trained GANs.

Diffusion models(Ho et al., 2020) are more powerful and stable than GANs and VAEs, and many works have succeeded in generating more realistic UAEs on complex and high-quality datasets. (Chen et al., 2023) is the first to investigate generating UAEs with diffusion models, it adds small adversarial perturbations to each latent code of the generation process and removes unnecessary noise via diffusion models to generate natural UAEs, it also leverages the information of original images to preserve semantic important objects. (Dai et al., 2025) generates realistic UAEs by using the gradient of defending classifiers to guide the latent code during each generation step. (Chen et al., 2024a) generates imperceptible and transferable UAEs by optimizing the attention map during the generation process of diffusion models. Furthermore, Diff-PGD (Xue et al., 2023) utilizes diffusion models to adapt adversarial examples generated by the PGD (Madry et al., 2017) method to align more closely with the real data distribution, resulting in more stealthy adversarial examples. Although it can be applied to various tasks, it is fundamentally based on the PGD method, a perturbation-based attack that depends on global noise patterns. Consequently, the naturalness and effectiveness of the adversarial examples remain unsatisfactory, despite the use of diffusion models to alleviate these patterns. To the best of our knowledge, all works so far consider using diffusion models as strong priors to enhance the generation of adver-

sarial samples only, and have not explored approximating the posterior distribution of UAEs yet.

## 2.2. Diffusion Models

Since (Ho et al., 2020) proposes denoising diffusion probabilistic models(DDPMs) and show their superiority in synthesizing high-quality and high-diversity images, the application range of diffusion models is becoming increasingly broad, such as image synthesis(Rombach et al., 2022)(Saharia et al., 2023)(Zhang et al., 2023), time series prediction(Tashiro et al., 2021)(Rasul et al., 2021), video synthesis(Harvey et al., 2022)(Ho et al., 2022), point cloud completion(Lyu et al., 2021)(Zhou et al., 2021), adversarial perturbations purification(**?**), etc.

DDPM is defined as a Markov chain comprising T forward diffusion steps, $x_{1:T}$, which convert an original image x into pure Gaussian noise and a series of Gaussian transitions comprising T reverse generation steps, $x_{T:1}$, which generate high-quality images with pure Gaussian noise input. In each forward diffusion step $t \in [1 : T]$, Gaussian noise is iteratively added to each latent code $x_t$ according to a monotonically increasing noise schedule $\beta_{1:T}$. Specifically,

$$q(x_t|x_{t-1}) = N(x_t; \sqrt{1-\beta_t}x_{t-1}, \beta_t\mathbf{I}) \qquad (1)$$

The reverse generation process, which begins with sampling $x_T$ from Gaussian distribution, generates each latent code $x_{t-1}$ by removing Gaussian noise from previous latent code $x_t$ and finally generates natural-like data $x_0$:

$$p_\theta(x_{t-1}|x_t) = N(x_{t-1}; \mu_\theta(x_t, t), \Sigma_\theta(x_t, t)) \qquad (2)$$

In DDPMs, $\mu_\theta(x_t, t) = \frac{1}{\sqrt{\alpha_t}}(x_t - \frac{1-\alpha_t}{\sqrt{1-\bar{\alpha}_t}}\epsilon_\theta(x_t, t))$, $\Sigma_\theta(x_t, t) \approx \beta_t$. Here, $\alpha_t = 1 - \beta_t$, $\bar{\alpha}_t = \prod_{i=1}^{t} \alpha_i$, $\epsilon_\theta(x_t, t)$ is Gaussian noise estimated by model $\epsilon_\theta$.

(Nichol & Dhariwal, 2021) proposes an improved DDPM that learns the variance schedule to improve quality and efficiency. Denoising Diffusion Implicit Models(DDIMs)(Song

et al., 2020) use a non-markovian diffusion process to achieve a much faster sampling speed than DDPMs with the same training procedure, whose generation process is:

$$p_\theta(x_{t-1}|x_t) = N\left(x_{t-1}; \sqrt{\bar{\alpha}_{t-1}}x_0 \right.$$
$$\left. + \sqrt{1 - \bar{\alpha}_{t-1} - \delta_t^2} \frac{x_t - \sqrt{\bar{\alpha}_t}x_0}{\sqrt{1 - \bar{\alpha}_t}}, \delta_t^2\mathbf{I}\right) \quad (3)$$

Here, $\delta_t \in [0, \sqrt{\frac{1 - \bar{\alpha}_{t-1}}{1 - \bar{\alpha}_t}}\sqrt{1 - \frac{\bar{\alpha}_t}{\bar{\alpha}_{t-1}}}]$ is the standard deviation, when $\delta_t = 0$, the generation process is deterministic, when $\delta_t = \sqrt{\frac{1 - \bar{\alpha}_{t-1}}{1 - \bar{\alpha}_t}}\sqrt{1 - \frac{\bar{\alpha}_t}{\bar{\alpha}_{t-1}}}$, the generation process is the same as DDPMs.

In addition, efforts are also made to improve the quality of conditional image generation. (Dhariwal & Nichol, 2021) leverages a pre-trained noisy classifier to guide class-conditional image synthesis. (Liu et al., 2023b) then extends it to image- and text-based guidance. (Ho & Salimans, 2022) further improves classifier guidance to classifier-free guidance, which utilizes an internal latent classifier.

## 3. Method

As is shown in Figure 1, DiffAdvMAP is formed with two branches to deal with two main attacking scenarios: whether or not a reference image exists. Suppose a real image is given for reference. In that case, most high-level features of the real image will be destroyed by going through the diffusion process for $t^*$ steps to obtain the latent code $x_t$; if not, a noisy image $x_T$ will be sampled from the Standard Gaussian distribution, it will go through the original generation process for $T - t^*$ steps until the latent code $x_t^*$ is obtained, where $T$ is the total length of the generation process, $t^*$ is a hyperparameter. Then the UAE is generated similarly for both scenarios with a $t^*$-steps-long truncated generation process. It approximates the posterior distribution of latent code $\hat{x}_t$ given the previous latent code $\hat{x}_{t+1}$ under the adversarial and reconstruction constraints, and samples $\hat{x}_t$ from it iteratively until the final UAE is generated. Going through the truncated generation process, we can generate adversarial high-level features by sampling from the approximated posterior distribution to generate more natural UAEs faster.

### 3.1. Diffusion-Based Adversarial Maximum a Posterior

We extend and adjust the MAP method to the diffusion-based UAEs generation task. We develop our methods under the Bayesian framework, which uses the Bayes formula to derive the posterior distribution of the target data, and samples the target data from this distribution by maximizing the probability density function(PDF) of the posterior distribution. In this section, we first construct the generation problem of UAEs in the form of mathematical formulas, then we derive the posterior distribution of UAEs with the

prior distribution of real data learned by the diffusion model based on the formulas. Afterward, we infer the PDF of the posterior distribution as our objective function. Finally, we follow a greedy optimization procedure to find each adversarial latent code $\hat{x}_{T:0}$ that maximizes the objective function to generate the final UAEs.

#### 3.1.1. POSTERIOR DISTRIBUTION DERIVATION

Given an optional reference real image $x$, a ground truth label $y$, a diffusion model $G_\theta$, and a target classifier $F_\phi$, our goal is to utilize $G_\theta$ to generate adversarial examples $\hat{x}_0$ that can deviate the decision of $F_\phi$ from correct to wrong:

$$F_\phi(Attack(G_\theta; y; x \ if \ exists)) = F_\phi(\hat{x}_0) \neq y \quad (4)$$

Here, if $x$ exists, $\hat{x}_0$ must be semantically close to $x$, and $Attack(\cdot)$ is our attack algorithm. From CW attack(Carlini & Wagner, 2017), we can convert the goal of $F_\phi(\hat{x}_0) \neq y$ into the adversarial constraint:

$$C_1 : Z(\hat{x}_0)_y - max_{i \neq y}(Z(\hat{x}_0)_i) = c \quad (5)$$

Where $c \leq 0$ is the confidence level of fooling the classifier, $Z(\hat{x}_0)_i$ is the logit output of classifier $F_\phi$ at entry $i$ with $\hat{x}_0$ as input. For convenience, we denote the logit difference $Z(\cdot)_y - max_{i \neq y}(Z(\cdot)_i)$ as $l(\cdot)$.

When generating UAE with a reference image, the difference between the UAE and the reference image is specifically defined in different scenarios, we introduce a reconstruction constraint to control the content of the UAE:

$$C_2 : m \circ \Omega(\hat{x}_0) = m \circ \Omega(\tilde{x}) \quad (6)$$

Here, $\circ$ means element-wise multiplication and $m$ is the mask used to deal with different kinds of regeneration regions. Specifically, when UAEs are generated globally, $m$ is an identity matrix; when UAEs are generated in some specified regions, $m$ is the mask that covers such specified regions. Only the regions $m$ covers should be generated when generating regional UAEs. Function $\Omega(\cdot)$ is a customized function for generating UAEs in different scenarios. For generating image-similar UAEs, $\tilde{x}$ is the original reference image, $\Omega(x) = x$; for generating style UAEs, $\tilde{x}$ is an extra image that contains the target style, $\Omega(\cdot)$ computes the style score as (Gatys et al., 2016); as for generating color UAEs, $\tilde{x}$ is the reference image after changing color, $\Omega(\cdot)$ converts images from the RGB space into the LAB space.

As a result, given the adversarial constraint $C_1$ and the reconstruction constraint $C_2$, the posterior distribution of UAEs can be represented as $p_\theta(\hat{x}_0|C_1, C_2)$, and since the reverse generation process of DDIMs is a deterministic process that once the input Gaussian noise $\hat{X}_T$ is determined, the output $\hat{x}_0$ is uniquely determined, the generation problem boils down to determine an appropriate $\hat{X}_T$ based on the

following posterior distribution:

$$p_\theta(\hat{x}_T|C_1, C_2) \propto p_\theta(\hat{x}_T)p_\theta(C_1|\hat{x}_T)p_\theta(C_2|\hat{x}_T) \quad (7)$$

Then, since the generation process of the diffusion model is a Markov process that can be decomposed as:

$$p_\theta(\hat{x}_{0:T}|C_1, C_2) = p_\theta(\hat{x}_T|C_1, C_2) \prod_{t=1}^{T} p_\theta(\hat{x}_{t-1}|\hat{x}_t, C_1, C_2) \quad (8)$$

As a result, we derive the posterior distribution of each adversarial latent code as follows:

$$p_\theta(\hat{x}_{t-1}|\hat{x}_t, C_1, C_2) \propto p_\theta(\hat{x}_{t-1}|\hat{x}_t)p_\theta(C_1|\hat{x}_{t-1})p_\theta(C_2|\hat{x}_{t-1}) \quad (9)$$

As for the scenario that the reference image is not given, we are supposed to generate the UAE from noise, the generated UAE must contain the object that can be recognized as the predefined label $y$ but be misclassified as another label by the target classifier. So we utilize the conditional diffusion model to generate most low-level features of the target object, and then generate adversarial features with DiffAdvMAP. Please refer to Appendix B for more details.

3.1.2. INFERENCE OF OBJECTIVE FUNCTION

For image-similar UAEs, we propose the PDF of the posterior distribution of each adversarial latent code $\hat{x}_t$ ($t \in [T : 1]$) in equation (8), Appendix B shows detailed derivation.

The approximation of the log PDF of equation (7) is:

$$\begin{aligned}
&\log p'_\theta(\hat{x}_T|C_1, C_2) \\
&= -\frac{1}{2}||\hat{x}_T||_2^2 - \frac{1}{2\xi_{1T}'^2}||c - l(f_\theta^T(\hat{x}_T))||_2^2 \\
&\quad - \frac{1}{2\xi_{2T}'^2}||x - f_\theta^T(\hat{x}_T)||_2^2 + C'
\end{aligned} \quad (10)$$

Where $\xi_i'(i = 1, 2)$ is the standard deviation of distribution $p_\theta(C_i|\hat{x}_T)(i = 1, 2)$, $C'$ is the normalizing constant, $f_\theta^t(\cdot)$ is a one-step estimation used to approximate the final UAE $\hat{x}_0$ with intermediate latent code $\hat{x}_t$ to reduce the computing complexity. The one-step estimation is defined as:

$$\hat{x}_0 \approx f_\theta^t(\hat{x}_t) = \frac{\hat{x}_t - \sqrt{1 - \bar{\alpha}_t}\epsilon_\theta(\hat{x}_t, t)}{\sqrt{\bar{\alpha}_t}} \quad (11)$$

Then the log PDF of the posterior distribution of each adversarial latent code from equation (9) can be approximated:

$$\begin{aligned}
&\log p'_\theta(\hat{x}_{t-1}|\hat{x}_t, C_1, C_2) \\
&= -\frac{1}{2\delta_t^2}||\hat{x}_{t-1} - \hat{\mu}_t||_2^2 - \frac{1}{2\xi_{1t-1}'^2}||c - l(f_\theta^{t-1}(\hat{x}_{t-1}))||_2^2 \\
&\quad - \frac{1}{2\xi_{2t-1}'^2}||x - f_\theta^{t-1}(\hat{x}_{t-1})||_2^2 + C'
\end{aligned} \quad (12)$$

Here, as is shown in equation (3), in DDIMs

$$\hat{\mu}_t = \sqrt{\bar{\alpha}_{t-1}}f_\theta^t(\hat{x}_t) + \sqrt{1 - \bar{\alpha}_{t-1} - \delta_t^2}\frac{\hat{x}_t - \sqrt{\bar{\alpha}_t}f_\theta^t(\hat{x}_t)}{\sqrt{1 - \bar{\alpha}_t}} \quad (13)$$

We follow a greedy optimization procedure to find each latent code $\hat{x}_{0:T}$, which samples an $\hat{x}_T$ by maximizing equation (10), and then samples $\hat{x}_{t-1}$ given previous latent code $\hat{x}_t$ by maximizing equation (12). Note that, each latent code $\hat{x}_{t-1}$ in equation (12) is initialized by $\hat{\mu}_t$. As for the approximation error introduced by the one-step estimation, it will reduce gradually as $t$ reduces and close to zero at the last few steps of the generation process(Zhang et al., 2023), so it has little effect on the quality and effectiveness of UAEs.

### 3.2. Destruction and Construction Method

Since going through the whole generation process of the diffusion model is time-consuming and recent research on real image editing(Mokady et al., 2023)(Couairon et al., 2022)(Kwon & Ye, 2022) show that perturbations can be applied to high-level semantics without compromising image realism. And (Meng et al., 2021)(Chung et al., 2022) present that generating adversarial features on real images can be regarded as a special case of real image editing. So we integrate a destruction and construction method with our framework, which allows us to preserve most low-level features of the original images and generate adversarial high-level features while accelerating the generation process.

This method is used for obtaining an appropriate intermediate latent code $x_t$ via the diffusion model as follows:

$$x_t \sim \begin{cases} q(x_t|x_0), \ x_0 \ exists \\ p_\theta(x_T) \prod_{i=T}^{t+1} p_\theta(x_{i-1}|x_i), \ otherwise \end{cases} \quad (14)$$

Here, $x_0$ is a reference image, $q(\cdot)$ means the diffusion process, $p_\theta(\cdot)$ means the generation process. Then $\hat{x}_T$ in equation (10) is initialized by $x_t$, and DiffAdvMAP is performed with a truncated generation process to generate UAEs. By integrating this method, the generation speed is improved greatly. The pseudo-code is shown in Appendix C.

## 4. Experiments

In this section, we evaluate the effectiveness of our framework under the black-box settings. This section is organized according to various attack conditions: generating UAEs from noise, global image-similar UAEs generation, regional image-similar UAEs generation, and customized UAEs generation. We will evaluate the transferability and robustness against defense methods of our framework in the global image-similar UAEs generation and generating UAEs from noise part. We also conduct evaluations under the white-box setting, please refer to Appendix D for more details.

## 4.1. Experimental Settings

**Datasets and Metrics.** We evaluate the performance of our framework on the ImageNet-compatible dataset(Kurakin et al., 2018), consisting of 1,000 images from ImageNet's validation set. In our experiments, we only consider the resolution of $224 * 224 * 3$. We apply the FID(Heusel et al., 2017) and LPIPS(Zhang et al., 2018) as the image quality metrics for global image-similar UAEs generation, FID, TRES(Golestaneh et al., 2022) and HyperIQA(Su et al., 2020) for generating UAEs from noise. Note that the reference data for computing the FID score is from DiffAttack.

**Models.** We adopt the latent diffusion model(**?**) for generating UAEs from noise, and a pre-trained unconditional DDPM from(Dhariwal & Nichol, 2021) in other attacking conditions. We select Inception V3(Inv-v3) (Szegedy et al., 2016), MobileNet V2(Mob-V2)(Sandler et al., 2018), Resnet50(Res-50)(He et al., 2016) and Swin-B(Liu et al., 2021) as the surrogate models, and evaluate the transferability of UAEs against each other. In addition, we also take various defense methods into consideration and evaluate the robustness against them: preprocessing methods(DiffPure(Nie et al., 2022), R&P(Xie et al., 2017), and NRP(Naseer et al., 2020) ) and adversarially trained models (Adv-Inc-v3(Kurakin et al., 2018), Inc-$v3_{ens3}$, Inc-$v3_{ens4}$, and IncRes-$v2_{ens}$(Tramèr et al., 2017)).

**Baseline Attacks.** For generating UAEs from noise, we choose AdvDiff(Dai et al., 2025) as the baseline method; for Global Image-Similar UAEs Generation, we choose three classical unrestricted attack methods(cAdv(Bhattad et al., 2019), ReColorAdv(Laidlaw & Feizi, 2019), and NCF(Yuan et al., 2022) ), two diffusion-based attack methods(Diff-PGD(Xue et al., 2023), and DiffAttack(Chen et al., 2024a)). We don't consider ACA(Chen et al., 2024b) since the official code isn't offered and the method is similar to DiffAttack.

**Implementation Details.** We leverage the DDIM sampling for the generation process. The number of diffusion steps $T$ is respaced to 200, $t = 40$, $c = -30$ and the number of DiffAdvMAP iterations is set to $I = 10$ for generating UAEs from noise. For other attacking conditions, $T = 100$, $t = 20$, $c = -40$ and $I = 2$. We apply an adaptive learning rate with an initial value of $lr = 0.01$, $\xi'_i(i = 1, 2)$ in equation (10) is set to 0.1 for all settings. All experiments are done with a single RTX3090 GPU.

## 4.2. Generating UAEs From Noise

Generating UAEs from noise is important for generative model-based adversarial attack methods, attackers can generate an unlimited number of UAEs once such an algorithm is developed. This can not only pose a great security challenge to DNNs but also offer enough AEs for adversarial training, thus improving the robustness of DNNs. We gen-

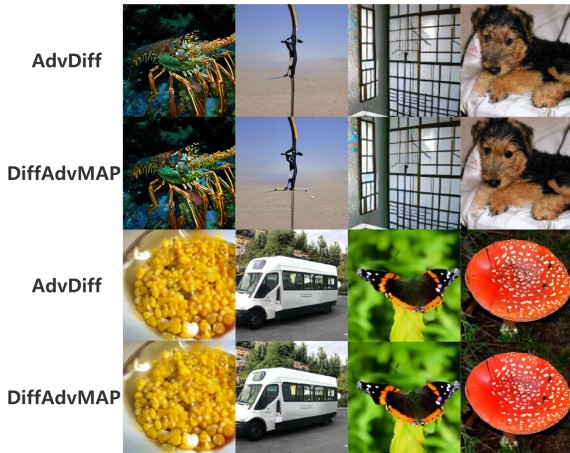

*Figure 2.* UAEs generated from noise using DiffAdvMAP and AdvDiff for attacking Resnet 50. We can see the eyes of the dog, the windows of the bus, and the body of the butterfly are more natural in UAEs generated by DiffAdvMAP.

erate one UAE for each class of the ImageNet dataset, the qualitative results are shown in Figure 2, and we can see that the UAEs generated by our method are more natural. We compare with AdvDiff quantitatively in Table 1 regarding attack success rate, transferability, and image quality. Our framework can generate UAEs with near $100\%$ white-box attack success rate while achieving better naturalness and transferability than the baseline method.

## 4.3. Global Image-Similar UAEs Generation

In this section, we conduct experiments regarding transferability, image quality assessment metrics: FID score, LPIPS metric, and defense robustness. Note that there is a trade-off between effectiveness and naturalness: larger perturbations are more likely to be robust against transfer and defense methods but can also diminish naturalness.

**Results on Normally Trained Models.** In this part, we evaluate the transferability between four normal DNNs, we select 3 classical unrestricted adversarial attack methods and two diffusion-based attack methods as our baseline. Table 2 shows the quantitative results of the white-box attack success rate and transferability. As we can see, our framework achieves near $100\%$ white-box attack success rate, which surpasses most baseline attacks. Meanwhile, in some model transfer experiments, DiffAttack and NCF achieves better transferability than our framework concerning the transfer attack success rate. However, our framework surpasses other baselines in all experiments, including the diffusion-based attack method Diff-PGD, which uses the same basic diffusion model as we do. Note that, our frame-

*Table 1.* The white-box attack success rate(%), transfer attack success rate (%), image quality metrics, as well as the run time(sec) of DiffAdvMAP and AdvDiff in the task of generating UAEs from noise. Since computing LPIPS score needs reference images, we replace it with blind image quality assessment metrics: TRES and HyperIQA.

| SURROGATE MODELS | ATTACK | DEFENDING MODELS | | | | FID(↓) | TRES(↑) | HYPERIQA(↑) | TIME |
| | | INC-V3 | RES-50 | MOB-V2 | SWIN-B | | | | |
|---|---|---|---|---|---|---|---|---|---|
| INC-V3 | ADVDIFF | **99.9** | 10.5 | 11.3 | 7.9 | 43.1 | 81.4 | 0.62 | **14.3** |
| | DIFFADVMAP(OURS) | 99.2 | **30.0** | **26.1** | **21.7** | 44.3 | **81.8** | **0.64** | 16.4 |
| RES-50 | ADVDIFF | 12.8 | **100.0** | 9.7 | 7.1 | 44.3 | 81.2 | 0.62 | - |
| | DIFFADVMAP(OURS) | **29.8** | **100.0** | **28.6** | **18.5** | **42.8** | **84.3** | **0.66** | - |
| MOB-V2 | ADVDIFF | 11.2 | 9.0 | **100.0** | 6.8 | 45.2 | 81.1 | 0.62 | - |
| | DIFFADVMAP(OURS) | **23.7** | **22.3** | 99.0 | **13.8** | **42.7** | **83.9** | **0.65** | - |
| SWIN-B | ADVDIFF | 13.6 | 11.5 | 12.3 | **98.7** | 43.7 | 81.5 | 0.63 | - |
| | DIFFADVMAP(OURS) | **24.6** | **25.2** | **23.9** | 97.3 | **43.0** | **83.2** | **0.65** | - |

work achieves near 50% attack success rate across various transfer models, which highlights the effectiveness of our framework in the transfer-based black-box attack. We also conduct ablation study in terms of each module and the adversarial confidence level $c$, please refer to Appendix E for more details.

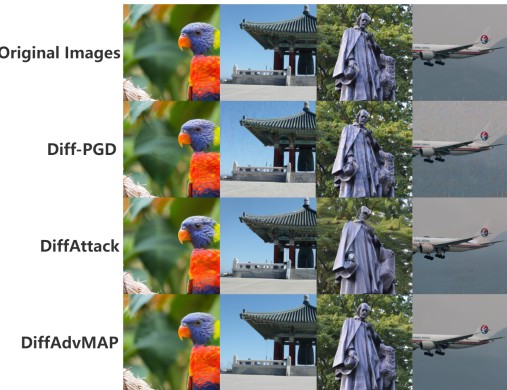

*Figure 3.* UAEs generated by the three diffusion-based methods, the surrogate model is Inception v3. UAEs generated by DiffAdvMAP don't have conspicuous noise patterns and preserve important low-level features. (eg: The face of the statue, the sign on the tail fin of the plane.)

Besides, we also evaluate the naturalness of adversarial examples as well as the time cost quantitatively in Table 2. Our framework achieves the best image quality among the attacks, it surpasses other diffusion-based methods including DiffAttack, which uses the stable diffusion model — a powerful model trained on a large set of high-quality data — which provides an optimal trade-off between naturalness and transferability. Compared with Diff-PGD, our framework demonstrates significantly better naturalness. In terms of the time cost, DiffAdvMAP still achieves a relatively low time cost. Figure 3 visualizes the adversarial examples generated by the three diffusion-based attack methods, providing a subjective perspective on their naturalness.

**Results on Denfense Robustness.** We also evaluate the robustness of our framework against three preprocessing defense methods and four adversarially trained models. After going through these strategies, we assess the effectiveness by calculating each attack method's white-box attack success rate. The results are shown in Table 3. We can see that DiffAdvMAP keeps a top-2 ranking in terms of the robustness against such defense strategies. The satisfactory robustness of DiffAdvMAP against such defense methods is due to the design of the adversarial constraint and the approach of sampling from the posterior distribution.

### 4.4. Regional Customized UAEs Generation

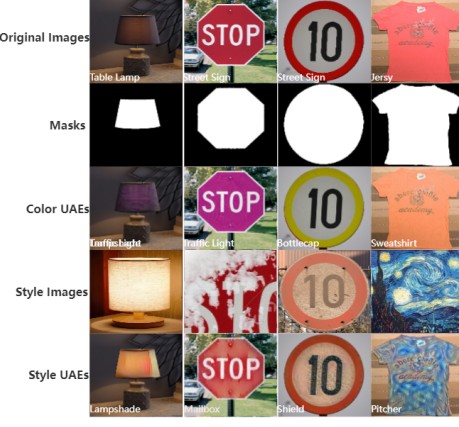

*Figure 4.* Qualitative results of regional color UAEs and style UAEs, the surrogate model is the Resnet50, predicted labels are in white.

In some scenarios, UAEs can only be generated by modifying some specified regions of the original images. Such UAEs can look similar to the original images or contain the same objects but differ in some attributes. In this section, we conduct experiments on generating customized UAEs: color UAEs generated by changing the color of reference images, and style UAEs generated by changing the style of reference images toward the target style. The qualitative

*Table 2.* The white-box attack success rate(%), transfer attack success rate (%), image quality metrics, as well as the run time(sec) of DiffAdvMAP and baseline methods in the task of generating global image-similar UAEs.

| SURROGATE MODELS | ATTACK | DEFENDING MODELS | | | | FID(↓) | LPIPS(↓) | TIME |
|---|---|---|---|---|---|---|---|---|
| | | INC-V3 | RES-50 | MOB-V2 | SWIN-B | | | |
| INC-V3 | CADV | 91.7 | 23.1 | 29.7 | 14.3 | 65.7 | 0.186 | 18.7 |
| | RECOLORADV | 98.4 | 31.6 | 39.3 | 15.0 | 63.4 | 0.154 | **3.86** |
| | NCF | 82.6 | **47.4** | **53.8** | 16.6 | 70.9 | 0.383 | 10.45 |
| | DIFF-PGD | 83.8 | 28.2 | 34.7 | 10.3 | 65.9 | 0.147 | 9.6 |
| | DIFFATTACK | 86.1 | 39.4 | 42.9 | 25.4 | 62.3 | **0.127** | 28.2 |
| | DIFFADVMAP(OURS) | **100.0** | 42.8 | 48.6 | 30.3 | 61.2 | 0.127 | 6.0 |
| RES-50 | CADV | 46.8 | 97.6 | 57.5 | 24.7 | 65.7 | 0.186 | - |
| | RECOLORADV | 47.9 | 99.2 | 63.8 | 28.1 | 63.4 | 0.154 | - |
| | NCF | 47.4 | 88.7 | 69.7 | 23.2 | 70.9 | 0.383 | - |
| | DIFF-PGD | 53.0 | 95.8 | 63.8 | 31.5 | 66.6 | 0.170 | - |
| | DIFFATTACK | **69.0** | 96.3 | 76.6 | 56.2 | 62.6 | 0.137 | - |
| | DIFFADVMAP(OURS) | 65.8 | **100.0** | **81.0** | **57.4** | 61.0 | 0.127 | - |
| MOB-V2 | CADV | 49.5 | 50.5 | 96.6 | 27.7 | 68.6 | 0.211 | - |
| | RECOLORADV | 48.7 | 40.4 | 99.8 | 30.1 | 63.3 | 0.157 | - |
| | NCF | 48.1 | 64.0 | 92.6 | 23.9 | 69.7 | 0.387 | - |
| | DIFF-PGD | 50.1 | 56.3 | 94.9 | 27.3 | 65.7 | 0.164 | - |
| | DIFFATTACK | **67.8** | 76.3 | 98.0 | 54.2 | 62.9 | 0.138 | - |
| | DIFFADVMAP(OURS) | 64.6 | **77.0** | **100.0** | **54.7** | 60.0 | 0.135 | - |
| SWIN-B | CADV | 43.2 | 40.9 | 46.1 | 98.4 | 67.4 | 0.191 | - |
| | RECOLORADV | 37.6 | 36.5 | 42.1 | 99.1 | 65.7 | 0.147 | - |
| | NCF | 39.5 | 50.5 | 55.1 | 63.1 | 65.5 | 0.346 | - |
| | DIFF-PGD | 41.2 | 46.6 | 53.1 | 94.7 | 70.6 | 0.189 | - |
| | DIFFATTACK | **57.7** | 56.6 | 58.4 | 90.1 | 65.5 | 0.138 | - |
| | DIFFADVMAP(OURS) | 55.6 | **56.9** | **63.5** | **99.1** | 64.9 | **0.125** | - |

*Table 3.* Evaluation of the robustness against three defense strategies. A higher white-box attack success rate(%) means better robustness. The surrogate model is Inception V3.

| ATTACKS | R&P | NRP | DIFFPURE | ADV-INC-V3 | INC-$v3_{ens3}$ | INC-$v3_{ens4}$ | INCRES-$v2_{ens}$ |
|---|---|---|---|---|---|---|---|
| CADV | 11.7 | 53.4 | 52.5 | 31.0 | 37.4 | 36.4 | 23.2 |
| RECOLORADV | 8.1 | 57.4 | 50.6 | 30.0 | 32.6 | 32.5 | 18.8 |
| NCF | 33.6 | 71.6 | 67.8 | 51.2 | 52.8 | 51.0 | 39.5 |
| DIFF-PGD | 30.0 | 57.3 | 64.7 | 31.9 | 37.0 | 34.7 | 19.1 |
| DIFFATTACK | 34.5 | 83.9 | 72.2 | **54.0** | 56.2 | 56.9 | 41.7 |
| DIFFADVMAP | **46.8** | **93.3** | **78.6** | 51.9 | **58.2** | **58.5** | **44.2** |

results of changing the color and style of specific objects in the original images are shown in Figure 4. We also conduct experiments on generating regional image-similar UAEs, which is a more broader perspective, please refer to Appendix F for more details.

## 5. Conclusion

In this paper, we introduce a flexible diffusion-based unrestricted adversarial attack framework, DiffAdvMAP. We generate natural UAEs by sampling adversarial latent code from the approximated posterior distribution of the UAEs. Near $100\%$ white-box attack success rate shows that our framework effectively defeats top-ranked robust models while keeping the naturalness of UAEs. In addition, our framework outperforms current SOTA with more naturalness and less time cost. DiffAdvMAP also achieves an

optimal trade-off between image naturalness, transferability, runtime, and defense robustness in the black-box setting, which makes it outperform most baseline attacks. Moreover, DiffAdvMAP is flexible enough to generate UAEs under various scenarios, making it more effective in various attack conditions, posing a significant challenge to DNNs.

## Impact Statement

This paper presents work to advance the Unrestricted adversarial attack. The community has discussed many potential societal consequences comprehensively, none of which we feel must be specifically highlighted here.

## Acknowledgement

This work was supported by the National Natural Science Foundation of China (NSFC) under Grant numbers 62171184, U23A20385, and 62273139. The authors sincerely acknowledge the foundation for their financial support, which made this research possible. The authors also acknowledge the constructive feedback of reviewers and the work of ICML'25 program and area chairs..

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

## A. Overview

Here is the overview of the appendix, we will first provide a detailed Bayesian derivation of the approximated posterior distribution as well as the derivation AdvMAP method in Appendix B. Then we put the pseudo-code of DiffAdvMAP and DiffAdvMAP-Region in Appendix C. The experimental results under the white-box setting will be evaluated in Appendix D. Appendix E shows the ablation study of each module and the super-parameters of DiffAdvMAP. Appendix F presents the experiments of Regional image-similar UAEs, we also visualize the qualitative results. Appendix G shows more qualitative results of UAEs generated by our framework.

## B. Detailed Bayesian Inference and Derivation of DiffAdvMAP

Given the input noise image $x_T \sim N(\mathbf{0}, \mathbf{I})$ of the diffusion model, the ground truth label $y$, the adversarial constraint $C_1$:

$$C_1 : Z(\hat{x}_0)_y - max_{i \neq y}(Z(\hat{x}_0)_i) = c \tag{15}$$

Where $c \leq 0$ is the confidence level of fooling the classifier, $Z(\hat{x}_0)_i$ is the logit output of classifier $F_\phi$ at entry $i$ with $\hat{x}_0$ as input. For convenience, we denote the logit difference $Z(\cdot)_y - max_{i \neq y}(Z(\cdot)_i)$ as $l(\cdot)$.

The reconstruction constraint $C_2$:

$$C_2 : m \circ \Omega(\hat{x}_0) = m \circ \Omega(\tilde{x}) \tag{16}$$

Here, $\circ$ means element-wise multiplication and $m$ is the mask used to deal with different kinds of regeneration regions. Specifically, when UAEs are generated globally, $m$ is an identity matrix; when UAEs are generated in some specified regions, $m$ is the mask that covers such specified regions. Only the regions $m$ covers should be generated when generating regional UAEs. Function $\Omega(\cdot)$ is a customized function for generating UAEs in different scenarios. For generating image-similar UAEs, $\tilde{x}$ is the original reference image, $\Omega(x) = x$; for generating style UAEs, $\tilde{x}$ is an extra image that contains the target style, $\Omega(\cdot)$ computes the style score; as for generating color UAEs, $\tilde{x}$ is the reference image after changing color, $\Omega(\cdot)$ converts images from the RGB space into the LAB space.

The posterior distribution of image-similar UAEs and UAEs generated from noise can be derived as follows:

$$
\begin{aligned}
p_\theta(\hat{x}_T | C_1, C_2) &= \frac{p_\theta(\hat{x}_T, C_1, C_2)}{p_\theta(C_1, C_2)} \\
&= \frac{p_\theta(C_1, C_2 | \hat{x}_T) * p_\theta(\hat{x}_T)}{p_\theta(C_1) * p_\theta(C_2)} \\
&= \frac{p_\theta(C_1 | \hat{x}_T) * p_\theta(C_2 | \hat{x}_T) * p_\theta(\hat{x}_T)}{p_\theta(C_1) * p_\theta(C_2)}
\end{aligned}
\tag{17}
$$

$$
\begin{aligned}
p_\theta(\hat{x}_T | y, C_1) &= \frac{p_\theta(\hat{x}_T, y, C_1)}{p_\theta(y, C_1)} \\
&= \frac{p_\theta(y, C_1 | \hat{x}_T) * p_\theta(\hat{x}_T)}{p_\theta(y) * p_\theta(C_1)} \\
&= \frac{p_\theta(y | \hat{x}_T) * p_\theta(C_1 | \hat{x}_T) * p_\theta(\hat{x}_T)}{p_\theta(y) * p_\theta(C_1)} \\
&= \frac{\frac{p_\theta(\hat{x}_T) * p_\theta(y)}{p_\theta(\hat{x}_T)} * p_\theta(C_1 | \hat{x}_T) * p_\theta(\hat{x}_T)}{p_\theta(y) * p_\theta(C_1)} \\
&= \frac{p_\theta(\hat{x}_T) * p_\theta(C_1 | \hat{x}_T)}{p_\theta(C_1)}
\end{aligned}
\tag{18}
$$

The posterior distribution of latent code $\hat{x}_{t-1}$ given latent code $\hat{x}_t$ can be derived:

$$
\begin{aligned}
p_\theta(\hat{x}_{t-1} | \hat{x}_t, C_1, C_2) &= \frac{p_\theta(\hat{x}_{t-1} | \hat{x}_t) * p_\theta(\hat{x}_t) * p_\theta(C_1, C_2 | \hat{x}_{t-1}, \hat{x}_t)}{p_\theta(\hat{x}_t, C_1, C_2)} \\
&= \frac{p_\theta(\hat{x}_{t-1} | \hat{x}_t) * p_\theta(\hat{x}_t) * p_\theta(C_1 | \hat{x}_{t-1}, \hat{x}_t) * p_\theta(C_2 | \hat{x}_{t-1}, \hat{x}_t)}{p_\theta(C_1 | \hat{x}_t) * p_\theta(C_2 | \hat{x}_t) * p_\theta(\hat{x}_t)} \\
&= \frac{p_\theta(\hat{x}_{t-1} | \hat{x}_t) * p_\theta(C_1 | \hat{x}_{t-1}) * p_\theta(C_2 | \hat{x}_{t-1})}{p_\theta(C_1 | \hat{x}_t) p_\theta(C_2 | \hat{x}_t)}
\end{aligned}
\tag{19}
$$

$$p_\theta(\hat{x}_{t-1}|\hat{x}_t, y, C_1) = \frac{p_\theta(\hat{x}_{t-1}|\hat{x}_t, y) * p_\theta(\hat{x}_t, y) * p_\theta(C_1|\hat{x}_{t-1}, \hat{x}_t, y)}{p_\theta(\hat{x}_t, y, C_1)}$$

$$= \frac{p_\theta(\hat{x}_{t-1}|\hat{x}_t, y) * p_\theta(y|\hat{x}_t) * p_\theta(\hat{x}_t) * p_\theta(C_1|\hat{x}_{t-1})}{p_\theta(C_1, y|\hat{x}_t) * p_\theta(\hat{x}_t)} \quad (20)$$

$$= \frac{p_\theta(\hat{x}_{t-1}|\hat{x}_t, y) * p_\theta(C_1|\hat{x}_{t-1})}{p_\theta(C_1|\hat{x}_t, y)}$$

Note that since we leverage a DDIM, whose generation process is deterministic, as a result, when $\hat{x}_T$ in equation (17)(18) and $\hat{x}_{t-1}$ in equation (19)(20) is given, adversarial goal $C_1$ and reconstruction constraint $C_2$ is independent. Then according to equations (15)(16) and (17), $p_\theta(\hat{x}_T)$ is a Gaussian distribution, and since when $\hat{x}_T$ is given, $\hat{x}_0$ is a deterministic function of $\hat{x}_T$, $p_\theta(l(\hat{x}_0) = c|x_T)$ and $p_\theta(m \circ \Omega(\hat{x}_0) = m \circ \Omega(\tilde{x}))$ follows Dirac delta function $\delta(\cdot)$, then $p_\theta(C_1|\hat{x}_T) = p_\theta(l(\hat{x}_0) = c|\hat{x}_T) = \delta(l(\hat{x}_0) = c)$, $p_\theta(C_2|\hat{x}_T) = p_\theta(m \circ \Omega(\hat{x}_0) = m \circ \Omega(\tilde{x})) = \delta(m \circ \Omega(\hat{x}_0) = m \circ \Omega(\tilde{x}))$, which means the probability density is infinite at $l(x_0) = c$, $m \circ \Omega(\hat{x}_0) = m \circ \Omega(\tilde{x})$, and 0 elsewhere. The Dirac delta function can be approximated by a Gaussian density function with zero variance. Therefore, if we take the logarithm to equation (17), we can approximate it as:

$$\log p_\theta(\hat{x}_T|C_1, C_2)$$
$$\approx -\frac{1}{2}||\hat{x}_T||_2^2 - \frac{1}{2\xi_1^2}||c - l(\hat{x}_0)||_2^2 - \frac{1}{2\xi_2^2}||m \circ \Omega(\hat{x}_0) - m \circ \Omega(\tilde{x})||_2^2 + C$$
$$\approx -\frac{1}{2}||\hat{x}_T||_2^2 - \frac{1}{2\xi_1^2}||c - l(G_\theta(\hat{x}_T))||_2^2 - \frac{1}{2\xi_2^2}||m \circ \Omega(G_\theta(\hat{x}_T)) - m \circ \Omega(\tilde{x})||_2^2 + C \quad (21)$$

where C is the normalizing constant, $\xi_1$ and $\xi_2$ are the standard deviation of the approximated Gaussian distribution of the Dirac delta functions respectively, when $\xi_1$ and $\xi_2$ approaches 0, the approximations go exact.

However, computing $G_\theta(\hat{x}_T)$ needs to go through the whole reverse generation process, which is time-consuming. As a result, we perform a one-step approximation of $\hat{x}_0$ for each step $t \in [T:1]$ in DDIM:

$$\hat{x}_0 \approx f_\theta^t(\hat{x}_t) = \frac{\hat{x}_t - \sqrt{1 - \bar{\alpha}_t}\epsilon_\theta(\hat{x}_t, t)}{\sqrt{\bar{\alpha}_t}} \quad (22)$$

Then the conditional distributions of $l(\hat{x}_0)$ and $\hat{x}_0$ given $\hat{x}_T$ can be approximated as Gaussian distributions centered around the one-step approximated value $f_\theta^T(x_T)$:

$$p_\theta'(l(\hat{x}_0)|\hat{x}_T) = N(l(\hat{x}_0); l(f_\theta^T(\hat{x}_T)), \xi_{1T}'^2\mathbf{I})$$
$$p_\theta'(m \circ \Omega(\hat{x}_0)|\hat{x}_T) = N(m \circ \Omega(\hat{x}_0); m \circ \Omega(f_\theta^T(\hat{x}_T)), \xi_{2T}'^2\mathbf{I}) \quad (23)$$

The approximation of log probability density computed from equation (17) and equation (18) are as follows:

$$\log p_\theta'(\hat{x}_T|C_1, C_2)$$
$$= \log(p_\theta(\hat{x}_T)) + \log(p_\theta'(l(\hat{x}_0) = c|\hat{x}_T)) + \log(p_\theta'(m \circ \Omega(\hat{x}_0) = m \circ \Omega(\tilde{x})|\hat{x}_T)) + C'$$
$$= -\frac{1}{2}||\hat{x}_T||_2^2 - \frac{1}{2\xi_{1T}'^2}||c - l(f_\theta^T(\hat{x}_T))||_2^2 - \frac{1}{2\xi_{2T}'^2}||m \circ \Omega(f_\theta^T(\hat{x}_T)) - m \circ \Omega(\tilde{x})||_2^2 + C' \quad (24)$$

$$\log p_\theta'(\hat{x}_T|y, C_1)$$
$$= \log(p_\theta(\hat{x}_T|y)) + \log(p_\theta'(l(\hat{x}_0) = c|\hat{x}_T)) + C'$$
$$= -\frac{1}{2}||\hat{x}_T||_2^2 - \frac{1}{2\xi_T'^2}||c - l(f_\theta^T(\hat{x}_T))||_2^2 + C' \quad (25)$$

$\xi_i'(i = 1, 2)$ is the standard deviation of approximated Gaussian distribution $p_\theta'(l(\hat{x}_0) = c|\hat{x}_T)$, which is different from $\xi_i(i = 1, 2)$ in equation (21), it should be large enough to capture the approximation error. Then since the generation process of UAEs under $C_1$ and $C_2$ can be decomposed as:

$$p_\theta'(\hat{x}_{0:T}|C_1, C_2) = p_\theta'(\hat{x}_T|C_1, C_2) \prod_{t=1}^T p_\theta'(\hat{x}_{t-1}|\hat{x}_t, C_1, C_2) \quad (26)$$

$$p'_\theta(\hat{x}_{0:T}|y, C_1) = p'_\theta(\hat{x}_T|y, C_1) \prod_{t=1}^{T} p'_\theta(\hat{x}_{t-1}|y, \hat{x}_t, C_1) \tag{27}$$

We can also approximate the intermediate conditional distributions given latent code $\hat{x}_t$ as:

$$p'_\theta(l(\hat{x}_0)|\hat{x}_t) = N(l(\hat{x}_0); l(f^t_\theta(\hat{x}_t)), \xi'^2_{1t}\mathbf{I})$$
$$p'_\theta(m \circ \Omega(\hat{x}_0)|\hat{x}_t) = N(m \circ \Omega(\hat{x}_0); m \circ \Omega(f^t_\theta(\hat{x}_t)), \xi'^2_{2t}\mathbf{I}) \tag{28}$$

Then the reverse generation process given equation (19)(20) can be computed as:

$$
\begin{aligned}
&\log p'_\theta(\hat{x}_{t-1}|\hat{x}_t, C_1, C_2) \\
&= \log(p_\theta(\hat{x}_{t-1}|\hat{x}_t)) + \log(p'_\theta(l(\hat{x}_0) = c|\hat{x}_{t-1}, \hat{x}_t)) + \log(p'_\theta(m \circ \Omega(\hat{x}_0) = m \circ \Omega(\tilde{x})|\hat{x}_{t-1}, \hat{x}_t)) + C' \\
&= log(p_\theta(\hat{x}_{t-1}|\hat{x}_t)) + \log(p'_\theta(l(\hat{x}_0) = c|\hat{x}_{t-1})) + \log(p'_\theta(m \circ \Omega(\hat{x}_0) = m \circ \Omega(\tilde{x})|\hat{x}_{t-1})) + C' \\
&= -\frac{1}{2\delta_t^2}||\hat{x}_{t-1} - \hat{\mu}_t||_2^2 - \frac{1}{2\xi'^2_{1t-1}}||c - l(f^{t-1}_\theta(\hat{x}_{t-1}))||_2^2 - \frac{1}{2\xi'^2_{2t-1}}||m \circ \Omega(f^{t-1}_\theta(\hat{x}_{t-1})) - m \circ \Omega(\tilde{x})||_2^2 + C'
\end{aligned}
\tag{29}
$$

$$
\begin{aligned}
&\log p'_\theta(\hat{x}_{t-1}|\hat{x}_t, y, C_1) \\
&= \log(p_\theta(\hat{x}_{t-1}|\hat{x}_t)) + \log(p'_\theta(l(\hat{x}_0) = c|\hat{x}_{t-1}, \hat{x}_t, y)) + C' \\
&= log(p_\theta(\hat{x}_{t-1}|\hat{x}_t)) + \log(p'_\theta(l(\hat{x}_0) = c|\hat{x}_{t-1})) + C' \\
&= -\frac{1}{2\delta_t^2}||\hat{x}_{t-1} - \hat{\mu}_t||_2^2 - \frac{1}{2\xi'^2_{t-1}}||c - l(f^{t-1}_\theta(\hat{x}_{t-1}))||_2^2 + C'
\end{aligned}
\tag{30}
$$

note that as is shown in equation (3), in DDIMs,

$$
\begin{aligned}
\hat{\mu}_t &= \sqrt{\bar{\alpha}_{t-1}}\hat{x}_0 + \sqrt{1 - \bar{\alpha}_{t-1} - \delta_t^2}\frac{\hat{x}_t - \sqrt{\bar{\alpha}_t}\hat{x}_0}{\sqrt{1 - \bar{\alpha}_t}} \\
&= \sqrt{\bar{\alpha}_{t-1}}f^t_\theta(\hat{x}_t) + \sqrt{1 - \bar{\alpha}_{t-1} - \delta_t^2}\frac{\hat{x}_t - \sqrt{\bar{\alpha}_t}f^t_\theta(\hat{x}_t)}{\sqrt{1 - \bar{\alpha}_t}}
\end{aligned}
\tag{31}
$$

## C. Pseudo-code

The pseudo-code of DiffAdvMAP and DiffAdvMAP-Region is shown in Alg.1 and Alg. 2 respectively.

---

**Algorithm 1** DiffAdvMAP

---

**Input:** optional reference image $x$, ground truth label $y$, diffusion model $\epsilon_\theta$, target classifier $F_\phi$, forward diffusion steps $t^*$, random Gaussian noise $\epsilon$, noise schedule $\beta_{1:T}$, MAP iterations $I$, MAP learning rate $lr$, adversarial confidence level $c$

**if** $x\ exists$ **then**
    $x_{t^*} \leftarrow \sqrt{\bar{\alpha}_{t^*}}x + (1 - \bar{\alpha}_{t^*})\epsilon$
**else**
    **for** $t = T$ **to** $t^* + 1$ **do**
        $x_{t-1} = \sqrt{\bar{\alpha}_{t-1}}f_\theta^t(x_t) + \sqrt{1 - \bar{\alpha}_{t-1}}\frac{x_t - \sqrt{\bar{\alpha}_t}f_\theta^t(x_t)}{\sqrt{1 - \bar{\alpha}_t}}$
    **end for**
**end if**
$\hat{x}_{t^*} = x_{t^*}$
**for** $i = 0$ **to** $I - 1$ **do**
    $\hat{x}_{t^*} = \hat{x}_{t^*} + lr * \nabla(\log(p'_\theta(\hat{x}_{t^*}|C_1, C_2)))$
    **if** $\arg\max F_\phi(f_\theta^{t^*}(\hat{x}_{t^*})) \neq y$ **then**
        **break**
    **end if**
**end for**
**for** $t = t^*$ **to** $1$ **do**
    $\hat{\mu}_t = \sqrt{\bar{\alpha}_{t-1}}f_\theta^t(\hat{x}_t) + \sqrt{1 - \bar{\alpha}_{t-1}}\frac{\hat{x}_t - \sqrt{\bar{\alpha}_t}f_\theta^t(\hat{x}_t)}{\sqrt{1 - \bar{\alpha}_t}}$
    $\hat{x}_{t-1} = \hat{\mu}_t$
    **for** $i = 0$ **to** $I - 1$ **do**
        $\hat{x}_{t-1} = \hat{x}_{t-1} + lr * \nabla(\log(p'_\theta(\hat{x}_{t-1}|\hat{x}_t, C_1, C_2))$
        **if** $\arg\max F_\phi(f_\theta^{t-1}(\hat{x}_{t-1})) \neq y$ **then**
            **break**
        **else**
            lr=lr*2
        **end if**
    **end for**
**end for**
**return** $\hat{x}_0$

---

**Algorithm 2** DiffAdvMAP-Region

---

**Input:** reference image $x$, mask $m$, ground truth label $y$, diffusion model $\epsilon_\theta$, target classifier $F_\phi$, forward diffusion steps $t^*$, random Gaussian noise $\epsilon$, noise schedule $\beta_{1:T}$, MAP iterations $I$, MAP learning rate $lr$, adversarial confidence level $c$

$x_{t^*} \leftarrow \sqrt{\bar{\alpha}_{t^*}}x + (1 - \bar{\alpha}_{t^*})\epsilon$
$\hat{x}_{t^*} = m \circ x_{t^*} + (1 - m) \circ x$
**for** $i = 0$ **to** $I - 1$ **do**
    $\hat{x}_0 = m \circ f_\theta^{t^*}(\hat{x}_{t^*}) + (1 - m) \circ x$
    $\hat{x}_{t^*} = \hat{x}_{t^*} + lr * \nabla(\log(p'_\theta(\hat{x}_{t^*}|C_1, C_2)))$
    **if** $\arg\max F_\phi(\hat{x}_0) \neq y$ **then**
        **break**
    **end if**
**end for**
**for** $t = t^*$ **to** $1$ **do**
    $\hat{\mu}_t = \sqrt{\bar{\alpha}_{t-1}}\hat{x}_t + \sqrt{1 - \bar{\alpha}_{t-1}}\frac{\hat{x}_t - \sqrt{\bar{\alpha}_t}\tilde{x}_0}{\sqrt{1 - \bar{\alpha}_t}}$
    $\hat{x}_{t-1} = m \circ \hat{\mu}_t + (1 - m) \circ x$
    **for** $i = 0$ **to** $I - 1$ **do**
        $\hat{x}_0 = m \circ f_\theta^{t-1}(\hat{x}_{t-1}) + (1 - m) \circ x$
        $\hat{x}_{t-1} = \hat{x}_{t-1} + lr * \nabla(\log(p'_\theta(\hat{x}_{t-1}|\hat{x}_t, C_1, C_2)))$
        **if** $\arg\max F_\phi(\tilde{x}_0) \neq y$ **then**
            **break**
        **else**
            lr=lr*2
        **end if**
    **end for**
**end for**
**return** $\hat{x}_0$

---

## D. UAEs Under the White-box Setting

In this part, we compare the naturalness and effectiveness of our framework in terms of image quality and attack success rate against both normal models and robust models. We select the state-of-the-art white-box diffusion-based unrestricted adversarial attack method: AdvDiffuser (Chen et al., 2023) as the baseline method. We evaluate the performance of our framework on the identical dataset as AdvDiffuser, it's a subset of the ImageNet test set which contains 1000 randomly selected images, 1 image for each class. We apply the FID score, LPIPS score, and SSIM metric to evaluate the quality of UAEs. As for the classifiers, we select a normally trained Resnet50 (He et al., 2016) as the baseline model, and three robust models from the RobustBench leaderboard (Croce et al., 2020) to evaluate the effectiveness of our framework against robust models: a Robust Resnet50 (Salman et al., 2020) B, a Robust Wide-Resnet50-2 (Salman et al., 2020) A (these two are current most robust convolutional networks), and an adversarially trained Resnet50 with the PGD attack (Engstrom et al., 2019). We also conduct experiments on vision transformer-based classifiers: a normally trained vit-b (Dosovitskiy, 2020), a Beit (Bao et al., 2021), and a robust vit-b (Singh et al., 2023) from RobustBench leaderboard. Note that since the authors of AdvDiffuser didn't offer their code, we compare our method with results proposed in their paper, so we also compare with AEs generated from Diff-PGD(Xue et al., 2023) with our reimplementation. We respace the number of diffusion steps from $T = 1000$ to $T = 400$ and set the forward diffusion step of DiffAdvMAP to $t = 3$. The adversarial confidence level is set to $c = -10$.

As is depicted in Figure 5, AEs generated by Diff-PGD contain obvious noise patterns when attacking robust models. Meanwhile, though UAEs generated by AdvDiffuser look natural, they change too many low-level features, taking unnatural features to UAEs when compared with the original images, for example, the bird's beak in the first column is almost gone. As for our framework, we leverage the prior knowledge of natural data to generate high-level adversarial features instead, which makes UAEs look more natural than AdvDiffuser and Diff-PGD.

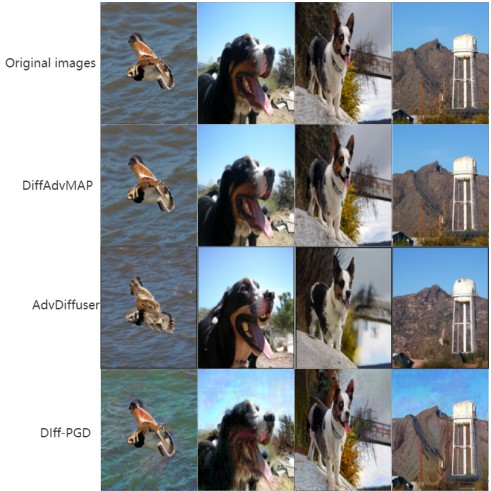

*Figure 5.* Image-similar UAEs generated by DiffAdvMAP, AdvDiffuser (Chen et al., 2023) and Diff-PGD (Xue et al., 2023). The defending model is the robust Wide-Resnet50-2 from (Salman et al., 2020).

We also conduct quantitative experiments to evaluate the effectiveness and naturalness of our framework. Table 4 presents the quantitative results of AdvDiffuser, Diff-PGD, and DiffAdvMAP respectively in terms of attack success rate, runtime for each sample, LPIPS score, SSIM metric, and FID score. As we can see, our framework achieves near $100\%$ white-box attack success rate against both normal models and robust models. Our framework also generates more natural UAEs, represented as much lower LPIPS, and FID scores. Meanwhile, the significantly reduced runtime addresses the inherent shortcoming of the slow generation speed of diffusion models, while still preserving the naturalness of UAEs.

We also amplify and compare the perturbations added by each method in Figure 6. We can observe that in UAEs generated by our framework, perturbations are tiny and coherent with class-specific semantics, so the prominence in areas with low information is greatly reduced. Our observations show that sampling UAEs from the posterior distribution of UAEs effectively improves their naturalness while maintaining a high attack success rate. Moreover, the use of the destruction

and construction method allows us to regenerate adversarial high-level features through a few generation steps, which improves the naturalness and generation speed further. In contrast, though AdvDiffuser achieves a better SSIM metric than our framework by introducing the information of original images in each step, it generates UAEs from the very beginning of the generation process and perturbs each latent code step by step, which suffers from diffusion models' slow generation speed. It also changes too many low-level features completely and only leverages the diffusion model to purify unnecessary perturbations, bringing strangeness and unnaturalness to UAEs(e.g. the claw of the chihuahua and the hog's nose and mouth.), and thus appears more noticeable and less imperceptible.

*Table 4.* Comparing global image-similar unrestricted attacks on ImageNet defending models, we also include the best-known robustness within $l_\infty = 4/255$ for each model.

| ATTACKER | ASR(%) | LPIPS(↓) | SSIM(↑) | FID(↓) | TIME |
|---|---|---|---|---|---|
| NORMAL RESNET50 (HE ET AL., 2016) | | | | | |
| $l_\infty$=4/255 | 100 | - | - | - | |
| DIFF-PGD | 98.7 | 0.180 | 0.82 | 57.61 | $\sim 10s$ |
| ADVDIFFUSER | 100 | 0.03 | **0.99** | 20.9 | $\sim 90s$ |
| DIFFADVMAP | **100** | **0.006** | 0.97 | **6.83** | $\sim$ **4**$s$ |
| ROBUST WIDE-RESNET50-2 FROM (SALMAN ET AL., 2020) | | | | | |
| $l_\infty$=4/255 | 61.9 | - | - | - | |
| DIFF-PGD | 88.6 | 0.201 | 0.82 | 81.54 | - |
| ADVDIFFUSER | **99.5** | 0.05 | 0.97 | 26.7 | - |
| DIFFADVMAP | 99.3 | **0.011** | **0.97** | **12.75** | - |
| ROBUST RESNET50 FROM (SALMAN ET AL., 2020) | | | | | |
| $l_\infty$=4/255 | 65.1 | - | - | - | |
| DIFF-PGD | 90.5 | 0.203 | 0.82 | 87.03 | - |
| ADVDIFFUSER | 99.8 | 0.05 | 0.97 | 27.2 | - |
| DIFFADVMAP | **99.8** | **0.007** | **0.97** | **9.95** | - |
| ROBUST RESNET50 FROM (ENGSTROM ET AL., 2019) | | | | | |
| $l_\infty$=4/255 | 70.8 | - | - | - | |
| DIFF-PGD | 91.5 | 0.21 | 0.80 | 89.26 | - |
| ADVDIFFUSER | 99.4 | 0.05 | **0.98** | 25.9 | - |
| DIFFADVMAP | **99.4** | **0.012** | 0.97 | **13.20** | - |
| NORMAL BEIT (DOSOVITSKIY, 2020) | | | | | |
| $l_\infty$=4/255 | 100 | - | - | - | |
| DIFF-PGD | 98.3 | 0.161 | 0.82 | 39.02 | - |
| ADVDIFFUSER | - | - | - | - | - |
| DIFFADVMAP | **100** | **0.006** | **0.97** | **3.87** | - |
| NORMAL VIT-B (BAO ET AL., 2021) | | | | | |
| $l_\infty$=4/255 | 100 | - | - | - | |
| DIFF-PGD | 92.3 | 0.182 | 0.82 | 43.96 | - |
| ADVDIFFUSER | - | - | - | - | - |
| DIFFADVMAP | **100** | **0.015** | **0.97** | **7.64** | - |
| ROBUST VIT-B FROM (SINGH ET AL., 2023) | | | | | |
| $l_\infty$=4/255 | 45.3 | - | - | - | |
| DIFF-PGD | 72.1 | 0.200 | 0.82 | 67.67 | - |
| ADVDIFFUSER | - | - | - | - | - |
| DIFFADVMAP | **93.8** | **0.015** | **0.97** | **12.41** | - |

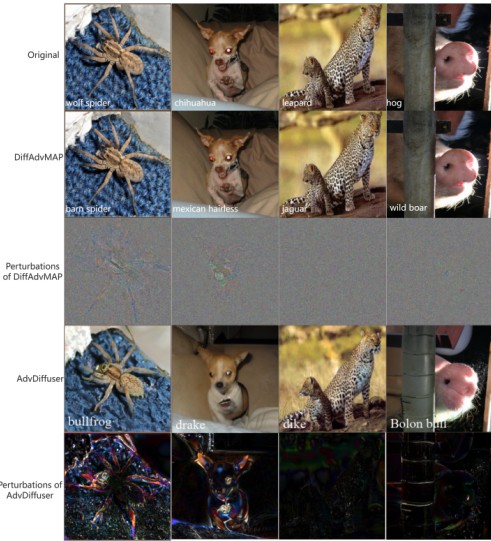

*Figure 6.* Perturbations generated by DiffAdvMAP and AdvDiffuser respectively. The defending model is a robust Wide-Resnet50-2 from (Salman et al., 2020). The predicted labels are shown in white.

## E. Ablation Study of Global Image-Similar UAEs Generation

In this section, we use Inception V3 as the surrogate model and analyze the effect of the reconstruction constraint and the forward diffusion step $t$ towards the UAEs in terms of white-box attack success rate, naturalness, transferability, and generate time. The results are reported in Table 5. As we can see, the destruction and reconstruction module makes great contribution towards improving the run time, but it is somewhat detrimental to the image quality; the adversarial constraint ensures the near 100% white-box attack success rate and high transferability, but it will reduce the image quality of UAEs; the reconstruction constraint in generating image-similar UAEs can help preserve important semantics of original images, thus improving the image quality, though it will reduce the transferabiltiy to some extent. In short, we find a better trade-off between transferability, white-box attack success rate and image quality in our DiffAdvMAP framework, and outperform other baseline attacks. We also make an ablation study of adversarial confidence level $c$, the results are shown in Table 6.

*Table 5.* Ablation Study of DiffAdvMAP in terms of each module on the Imagenet compatible dataset, the surrogate model is Inception V3. w/o means without the module in the framework.

| SETTINGS | WHITE-BOX ATTACK | NATURALNESS | | TRANSFERABILITY | | | TIME |
| --- | --- | --- | --- | --- | --- | --- | --- |
| | SUCCESS RATE | FID($\downarrow$) | LPIPS($\downarrow$) | RES-50 | MOB-V2 | SWIN-B | |
| W/O RECONSTRUCTION CONSTRAINT | 100.0 | 63.3 | 0.171 | 52.5 | 55.1 | 40.8 | 6.0 |
| W/O ADVERSARIAL CONSTRAINT | 25.0 | 58.1 | 0.056 | 11.4 | 16.2 | 5.7 | 6.0 |
| W/O DESTRUCTION AND RECONSTRUCTION | 100.0 | 61.8 | 0.116 | 35.9 | 45.2 | 24.4 | 44.5 |
| DIFFADVMAP(OURS) | 100.0 | 61.2 | 0.127 | 42.8 | 48.6 | 30.3 | 6.0 |

As we can see, the adversarial constraint ensures that our framework consistently achieves a near 100% white-box attack success rate, regardless of the adversarial confidence level. As the absolute value of the confidence level increases, the attack strength of the UAEs also increases, which is evident in their enhanced transferability. However, the image quality diminishes as the absolute value of $c$ rises, due to more prominent adversarial features, as discussed in the first paragraph of Section 4.3. Nonetheless, the experimental results demonstrate that our framework achieves a superior balance between naturalness and attack strength compared to baseline attacks.

*Table 6.* Ablation Study of DiffAdvMAP in terms of adversarial confidence level $c$ on the Imagenet compatible dataset, the surrogate model is Inception V3.

| ADVERSARIAL CONFIDENCE LEVEL $c$ | WHITE-BOX ATTACK | NATURALNESS | | TRANSFERABILITY | | |
|---|---|---|---|---|---|---|
| | SUCCESS RATE | FID($\downarrow$) | LPIPS($\downarrow$) | RES-50 | MOB-V2 | SWIN-B |
| -5 | 98.9 | 58.5 | 0.085 | 16.3 | 23.8 | 10.7 |
| -10 | 99.7 | 58.5 | 0.089 | 20.5 | 27.6 | 11.5 |
| -15 | 99.9 | 58.5 | 0.092 | 23.5 | 30.5 | 14.2 |
| -20 | 100.0 | 59.0 | 0.098 | 26.7 | 34.1 | 16.6 |
| -25 | 100.0 | 59.5 | 0.106 | 30.3 | 38.2 | 20.0 |
| -30 | 100.0 | 59.9 | 0.112 | 35.2 | 40.9 | 22.9 |
| -35 | 100.0 | 60.4 | 0.119 | 38.4 | 44.5 | 26.4 |
| -40 | 100.0 | 61.2 | 0.127 | 42.8 | 48.6 | 30.3 |

# F. Regional Image-Similar UAEs Generation

In this section, we leverage random square masks to specify the regions to be perturbed, generating regional image-similar UAEs. We compare the regional adversarial examples with Diff-PGD visually in Figure 4. We select an adversarially trained Resnet50 as the surrogate model. As we can see, in the specified regions, our framework can still generate natural adversarial features, while for Diff-PGD, strange textures and noise patterns exist in the image, for instance, in the first column, the white flower displays a strange red color; in the fourth column, the noise texture in the left bottom of the fig is quite obvious.

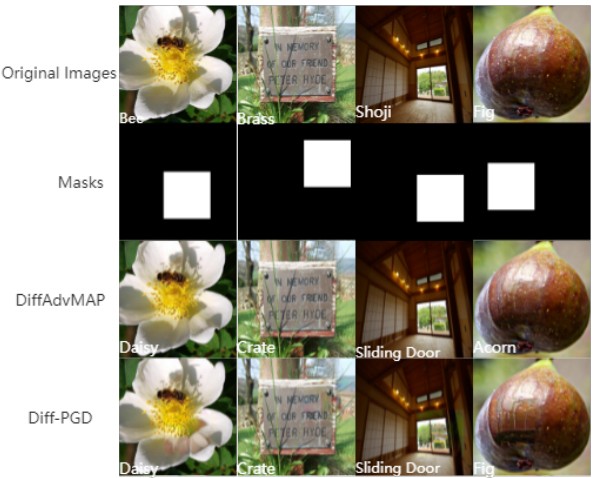

*Figure 7.* UAEs generated by DiffAdvMAP-Region with random square masks, the defending model is a robust Resnet50 from (Engstrom et al., 2019), predicted labels are in white.

We also introduce more qualitative results as well as the perturbations in Figure 8, the regional UAEs are generated against different robust models with two kinds of masks: random square masks and irregular masks generated by Grad-CAM(Selvaraju et al., 2017) method. For random square masks, perturbations become extremely significant, especially when most of the mask doesn't include semantic useful information, leading to unnatural adversarial examples or even failure. If combined with Grad-CAM to find out the region where the defending model extracts features to predict the ground truth label, the perturbations become coherent with semantics and thus less significant and perceptible.

We also compare our method with Diff-PGD quantitatively, as is shown in Table 7, the experiments are done under the white-box setting. As we can see, we achieve a much higher attack success rate against normal and robust models than Dif-PGD while maintaining a better image quality.

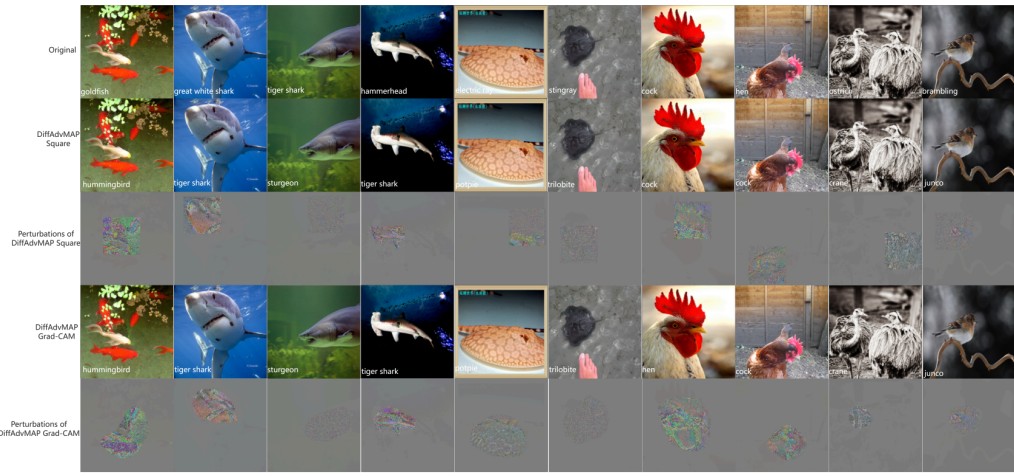

*Figure 8.* UAEs in specified regions against a robust Wide-Resnet50-2 from (Salman et al., 2020) with square masks and Grad-CAM masks respectively. We also include corresponding perturbations and predicted labels.

*Table 7.* Comparing Regional UAEs on ImageNet defending models.

| ATTACKER | ASR(%) | LPIPS | SSIM | FID |
|---|---|---|---|---|
| NORMAL RESNET50 | | | | |
| $l_\infty$=4/255 | 100 | - | - | - |
| DIFF-PGD | 89.7 | 0.03 | 0.97 | 24.53 |
| DIFFADVMAP | **99.8** | **0.01** | **0.99** | **7.05** |
| ROBUT WIDE-RESNET50-2 FROM (SALMAN ET AL., 2020) | | | | |
| $l_\infty$=4/255 | 61.9 | - | - | - |
| DIFF-PGD | 36.5 | 0.06 | 0.96 | 30.64 |
| DIFFADVMAP | **81.6** | **0.02** | **0.98** | **14.75** |
| ROBUT RESNET50 FROM (SALMAN ET AL., 2020) | | | | |
| $l_\infty$=4/255 | 65.1 | - | - | - |
| DIFF-PGD | 42.4 | 0.05 | 0.96 | 30.18 |
| DIFFADVMAP | **91.8** | **0.01** | **0.98** | **10.38** |
| ROBUT RESNET50 FROM (ENGSTROM ET AL., 2019) | | | | |
| $l_\infty$=4/255 | 70.8 | - | - | - |
| DIFF-PGD | 45.3 | 0.06 | 0.96 | 32.64 |
| DIFFADVMAP | **86.9** | **0.02** | **0.98** | **14.01** |

## G. More Qualitative Results of UAEs Generated by DiffAdvMAP

In this section, we will propose more qualitative results of various UAEs generated by DiffAdvMAP, as shown in Figures 9, 10, 11, and 12.

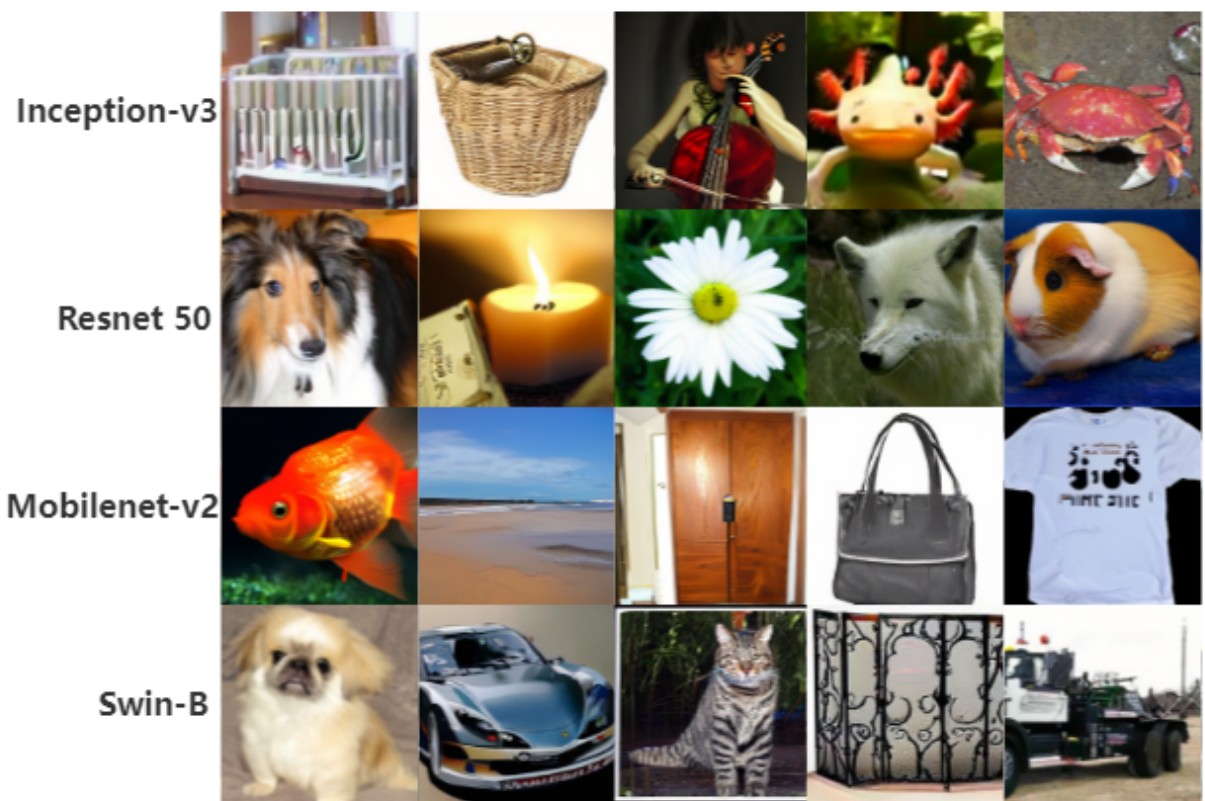

*Figure 9.* UAEs generated from noise with four normal DNNs as surrogate models.

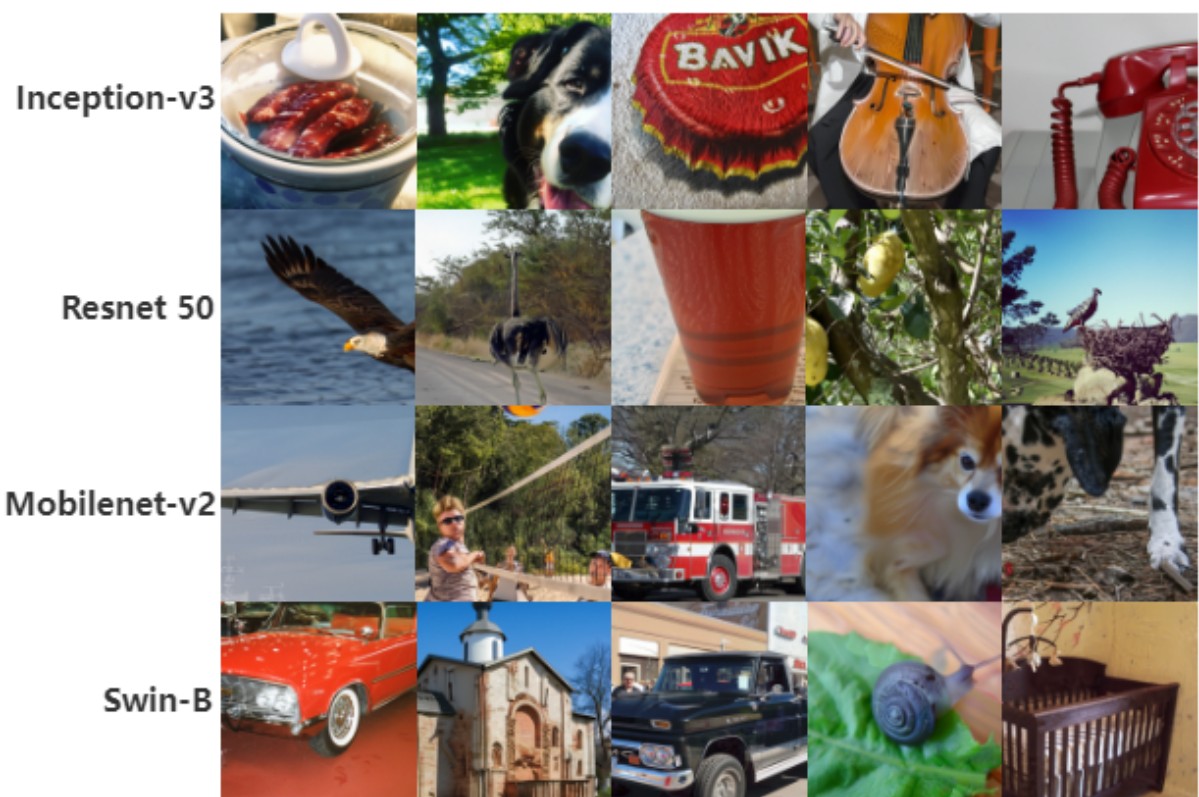

*Figure 10.* Global image-similar UAEs with four normal DNNs as surrogate models.

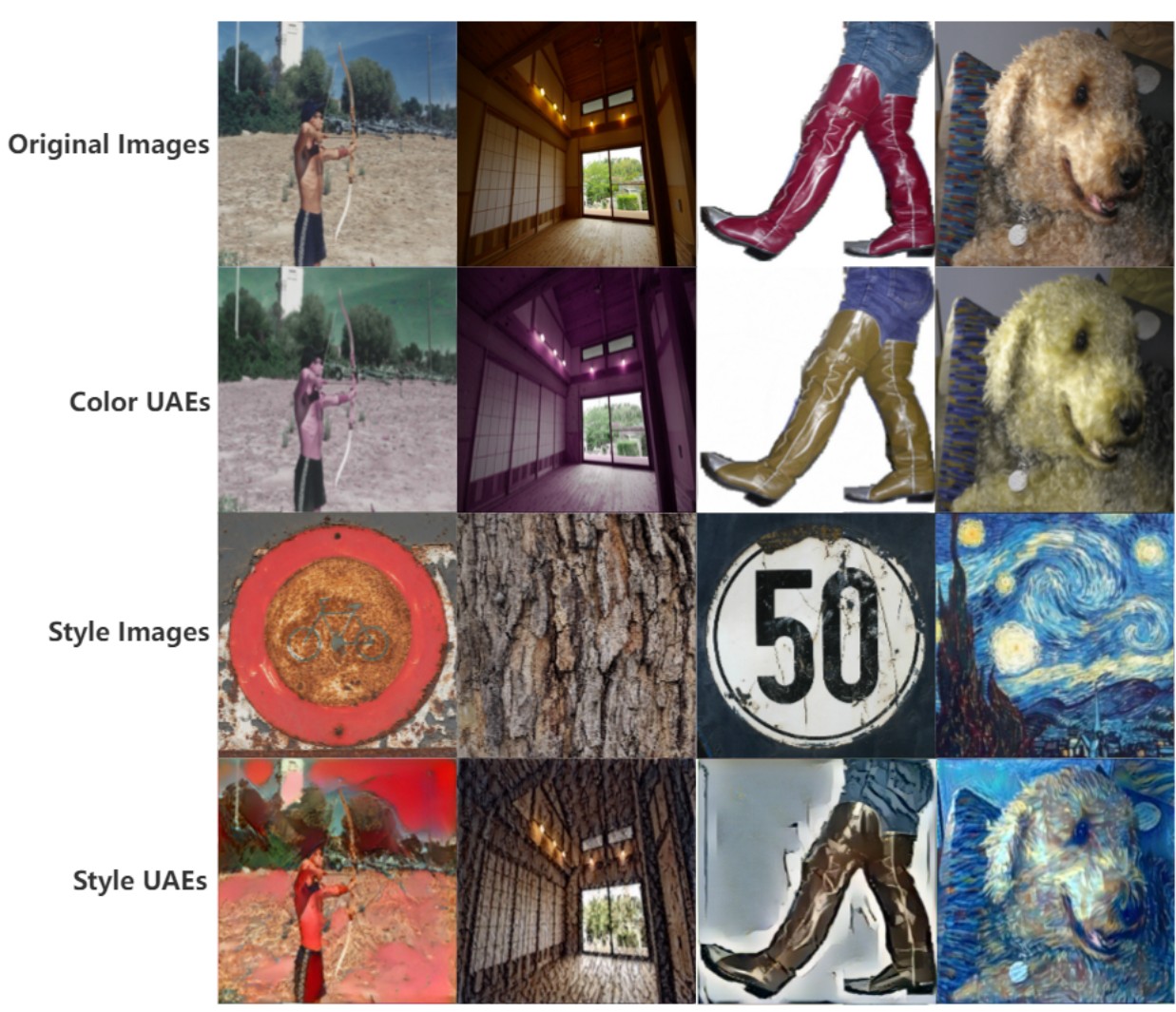

*Figure 11.* Global customized UAEs generated by DiffAdvMAP, the surrogate model is normal Resnet50.

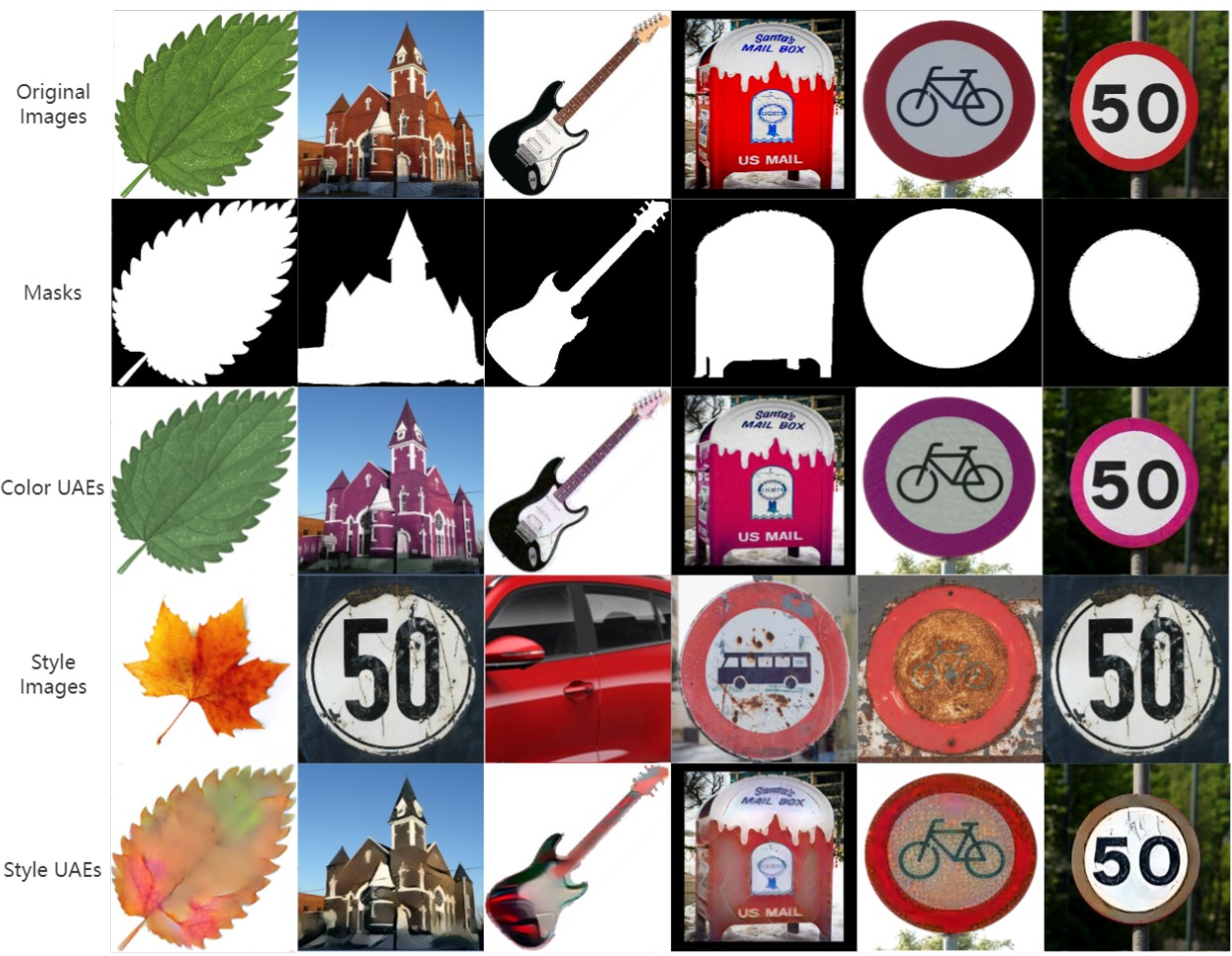

*Figure 12.* Regional customized UAEs generated by DiffAdvMAP, the surrogate model is normal Resnet50.

