# OpenReview forum: "DiffAdvMAP: Flexible Diffusion-Based Framework for Generating Natural Unrestricted Adversarial Examples"
_ICML.cc/2025/Conference — ICML 2025 poster_

### Official Review · Reviewer_4Qqs · 2025-03-07

**Overall Recommendation:** 3

**Summary:**

In this paper, the authors propose a flexible framework named DiffAdvMAP to facilitate the effective and natural generation of unrestricted adversarial examples (UAEs). The framework is based on the posterior distribution with two constraints of adversary and reconstruction, which is eventually optimized with a gradient method. Experiments validate the effectiveness, efficiency and naturalness of the proposed method.

**Claims And Evidence:**

Yes. The writing is clear.

**Essential References Not Discussed:**

AdvDiffuser published in ICCV2023 is suggested to be included as a baseline.

**Experimental Designs Or Analyses:**

The experiments are extensive with different settings and baselines. Ablations of important modules are conducted. Results are analyzed with sufficient visualizations.

**Methods And Evaluation Criteria:**

The two constraints adopted in the framework are well-motivated and straightforward to achieve high ASRs and flexibility under different settings. Evaluation on NIPS2017 with diverse metrics like FID, LPIPS in addition to ASR is acceptable.

**Other Comments Or Suggestions:**

* In line 36, a string of "0" is injected to the main text.
* The resolution of Figure 1 should be higher.

**Other Strengths And Weaknesses:**

Strength:
* The paper formulates the generation of UAEs from a Bayesian perspective, which is interesting.
* The method is well-motivated to combine two constraints into the framework.
* The experiments are extensive with sufficient analyses.

Weakness:
* The method eventually ends up with a loss function with three regularization terms, which is optimized simply with gradient methods. For the term corresponding to the adversarial constraint, it's different from previous methods only because of the utilization of CW attack, rather than original likelihood maximization. There should be more discussions about the practical difference between the proposed method and previous methods and the essential factors that make the method more effective.
* There is a phenomenon that the transfer ASRs for Sec. 4.2 are lower in general than those in Sec. 4.3. Is there any explanation?

**Questions For Authors:**

See comments above.

**Relation To Broader Scientific Literature:**

This paper provides a unified and flexible theoretical framework for the generation of UAEs with diffusion models.

**Theoretical Claims:**

The formulation and logic of the theoretical claims seems reasonable.

One minor question is about the last line of Eq. (20). The equality holds when $p(y|\hat{x}_t)p(C_1) = p(C_1,y|\hat{x}_t)$, which is only correct when $C_1$ is independent of $y$ and $\hat{x}_t$. This is clearly wrong given the definition of $C_1$.

---

> ### Author Rebuttal · Authors · 2025-04-01
>
> Thank you for your reviews and kindly reminder. According to your opinions, we add the theoretical explanation of the effetviness of DiffAdvMAP, and explain the reason for the difference of transfer ASRs in Sec. 4.2 and Sec. 4.3, the responses are as follows. We wish you can improve the score after reading them.
> 1. The essential factors that make the method more effective: As is illustrated in Section 3, we introduce an adversarial constraint to ensure the effectiveness of UAEs and a reconstruction constraint to control the content of generated UAEs, such two constraints are considered as two events of the posterior distribution of UAEs. Then the posterior distribution is approximated with the prior knowledge of real data learned by the diffusion model, unlike previous methods which only use diffusion models as strong denoisers to enhance the generation process, DiffAdvMAP generates adversarial features by sampling from the approximated posterior distribution, which make the adversarial features more coherent to the original features, thus improving the naturalness of UAEs. As a result, we have more space to improve effectiveness at the cost of sacrificing naturalness. Besides, in order to derive the probability density function(PDF) of the posterior distribution, the events should be deterministic events that are expressed as equations. In this situation, the objective function of CW attack can be converted into the form of an equation, on the right side of the equation is the adversarial confidence level $c$, once $c\le 0$, the adversarial example can attack the classifier successfully, as for the likelihood function, there’s no exact range indicates that the attack will succeed. As a result, we can control the strength of UAEs more effectively by controlling the value of confidence level c, such that we can achieve a better trade-off between the effectiveness and naturalness of UAEs. As is illustrated in Table 6 in Appendix E, the white-box attack success rate is nearly 100%, but the transfer ASR is not enough when the absolute value of the confidence level is small. When the absolute value of the confidence level increases, the transferability increases, and although the naturalness decreases, it is still better than other baselines.
> 2. The reason for transfer ASRs in Sec. 4.2 are different from those in Sec. 4.3: On the one hand, experiments in Section 4.2 and Section 4.3 undertake different tasks with different models: experiments in Section 4.2 generate UAEs from noise with the latent diffusion model[1] and experiments in Section 4.3 generate image-similar UAEs with real images as reference. The input and the generated UAEs are completely different. On the other hand, since there’s no real image for reference in the task of generating UAEs from noise, we employ a strong class guidance to ensure the naturalness of generated UAEs, then it becomes harder for DiffAdvMAP to counteract the strong class guidance to generate adversarial features, thus the effectiveness of generated UAEs decreases when maintianing their naturalness, which is reflected by the transfer attack success rate.
> 3. Including important paper: Since the authors of AdvDiffuser don’t release their code(though there’s a github repository proposed in the paper, the repository is empty), and it conducts the experiments under the white-box setting, which is different from the setting in our main paper, we have compared DiffAdvMAP with AdvDiffuser in Appendix D under the white-box setting, the evaluation metrics are all from their paper, please refer to it for more details, it’s obvious that the efficiency, effectiveness and naturalness of UAEs generated by our method is better than AdvDiffuser.
> 4. Typos and suggestions: We are sorry we made such typos, the denominator of equaiton(20) should be $p(C_1|x_t,y)$, but it has no impact on the subsequent derivation, we will correct the typos here and in line 36, increase the resolution of Figure in the revised version.

---

> > ### Comment · Reviewer_4Qqs · 2025-04-02
> >
> > Thanks for your thorough and detailed response. I will keep my rating for acceptance.

---

### Official Review · Reviewer_nVug · 2025-03-11

**Overall Recommendation:** 2

**Summary:**

This paper introduces Diffusion-based Adversarial Maximum a Posteriori (DiffAdvMAP), a framework that generates unrestricted UAEs by sampling from posterior distributions. It leverages a diffusion model to approximate the prior distribution of real data and incorporates adversarial and reconstruction constraints within a Bayesian framework to enhance both effectiveness and realism. Experimental results on ImageNet demonstrate that DiffAdvMAP achieves a superior trade-off between image quality, flexibility, and transferability compared to existing unrestricted adversarial attack methods.

**Claims And Evidence:**

The claimed technical contributions are overstated. The authors assert that their framework can generate UAEs under various attack conditions, including noise-based and global image-similar UAEs. However, as shown in Fig. 1, generating UAEs from noise simply involves using a pretrained diffusion model to produce a source image, then applying adversarial loss to an intermediate result—making it technically no different from global image-similar UAEs generation. Similarly, regional customized UAEs generation merely applies a mask and/or style loss to the source image, which is not a novel approach.

**Essential References Not Discussed:**

n/a

**Experimental Designs Or Analyses:**

n/a

**Methods And Evaluation Criteria:**

In Section 4, the authors claim that the evaluation is conducted under a black-box setting, which does not align with the titles of the subsequent tables.

Furthermore, the classification of UAE generation types (Sections 4.2 and 4.3) appears inconsistent with prior works, where AdvDiff and DiffAttack are typically considered the same type of method. Additionally, the authors fail to include AdvDiffuser [1], an important related work.

[1]AdvDiffuser: Natural Adversarial Example Synthesis with Diffusion Models

**Other Comments Or Suggestions:**

n/a

**Other Strengths And Weaknesses:**

n/a

**Questions For Authors:**

n/a

**Relation To Broader Scientific Literature:**

n/a

**Theoretical Claims:**

n/a

---

> ### Author Rebuttal · Authors · 2025-04-01
>
> We are sorry that maybe because our writing is not clear enough, you have some misunderstandings about the innovation of our paper. Our responses are as follows, and we hope you can improve the score after reading our responses and paper carefully.
> 1. DiffAdvMAP is a flexible framework that can be adapted to different attacking scenarios by sampling UAEs from their approximated posterior distribution under the adversarial constraint and reconstruction constraint. We make specific adaptations to each attacking scenario.\
> (1) Technical $\textbf{Difference}$ between generating UAEs from noises and image-similar UAEs:  When generating UAEs from noise, we generate most low-level semantics via the generation process and then generate high-level adversarial features; when generating image-similar UAEs, we destroy the high-level features of the input clean image with several diffusion steps, and regenerate high-level adversarial features, as is shown in Alg.1 in Appendix C. Besides, we leverage different diffusion models for such two tasks: the latent diffusion model and an unconditional diffusion model, we also build different objective functions for each task: we introduce the reconstruction constraint based on the reference image in the task of generating image-similar UAEs, while in the task of generating UAEs from noise, we replace the reconstruction constraint with class condition y, as is introduced in Section 3 and Appendix B. The experimental results in Section 4 also show that DiffAdvMAP can generate UAEs from noise and generate image-similar UAEs. \
> (2) DiffAdvMAP achieves $\textbf{innovations}$ in generating regional and customized unrestricted adversarial examples (UAEs): For regional UAEs, it abandons the traditional additive perturbation-based approach applied to masked areas. Instead, it destroys mask-covered regions and reconstructs adversarial features through posterior distribution sampling under dual constraints (adversarial and reconstruction), enabling generative reconstruction of adversarial patterns. For customized UAEs, it embeds specific reconstruction constraints (e.g., style/color consistency) into the posterior probability density function, replacing explicit style/color loss functions. This directly guides probability distribution shaping via reference images to ensure precise semantic alignment.
> 2. Table title is $\textbf{aligned}$ to section content: In section 4, we conduct experiments under the transfer-based black-box setting, the title of Table 2 ‘The white-box attack success rate(%), transfer attack success rate (%), image quality metrics, as well as the run time(sec) of DiffAdvMAP and baseline methods in the task of generating global image-similar UAEs’ means the white-box attack success rate(ASR) of attacking white-box known surrogate models, and transfer ASR of attacking black-box unkown classifiers is included in the table. When performing a transfer-based black-box attack, UAEs are generated with a known surrogate model, eg, Inception V3, then these UAEs are used to attack other unknown classifiers, eg, Mobilenet V2, Resnet50, and Swin-Transformer, and obtain the transfer attack success rate[1]. As a result, the evaluation is conducted under a black-box setting, and the content and the title of the tables in Section 4 are consistent with our claims.\
> [1] Chen Y, Liu W. A theory of transfer-based black-box attacks: Explanation and implications[J]. Advances in Neural Information Processing Systems, 2023, 36: 13887-13907.
> 3. Advdiff and DiffAttack are $\textbf{different}$: According to the paper and official code of AdvDiff and DiffAttack, AdvDiff leverages the latent diffusion model to generate UAEs from noise, its inputs are random Gaussian noise images, it designs two adversarial guidance techniques to conduct adversarial sampling in the reverse generation process of diffusion models; DiffAttack leverages the stable diffusion model to generate image-similar UAEs, its inputs are clean images, it computes the cross attention map and self attention map of each latent code as the objective fucntion and optimizes the objective function during the generation. As a result, though AdvDiff and DiffAttack are all diffusion-based methods, the principles and corresponding tasks are different, so we compare the performance of our method in different tasks with them in different sections.
> 4. Include important paper: Since the authors of AdvDiffuser did’t release their code(though there’s a github repository proposed in the paper, the repository is empty), and it conducts the experiments under the white-box setting, which is different from the setting in our main paper, we have compared DiffAdvMAP with AdvDiffuser in Appendix D under the white-box setting, the evaluation metrics are all from their paper, please refer to it for more details, it’s obvious that the efficiency, effectiveness and naturalness of UAEs generated by our method is better than AdvDiffuser.

---

> > ### Comment · Reviewer_nVug · 2025-04-07
> >
> > Thank you for your response. The rebuttal addressed some of my concerns; however, my main concern regarding the necessity of separating the generation of UAEs from noise-based and image-similar cases remains unaddressed. The only distinction appears to be how the 'low-level semantics' images are obtained. Since the proposed DiffAdvMAP block is applied after generating these images, I don't see a strong justification for treating these conditions separately or presenting this distinction as a key contribution. Notably, prior work [1] has also compared AdvDiff and DiffAttack within the same evaluation table. Given this, I will raise my score to a weak reject.
> >
> > [1] AdvDiff: Generating Unrestricted Adversarial Examples using Diffusion Models

---

> > > ### Author Response · Authors · 2025-04-07
> > >
> > > We sincerely appreciate your careful consideration of our rebuttal and thoughtful evaluation adjustment. To facilitate a deeper understanding of this research, we have prepared a detailed explanation that addresses each point of concern. We would be pleased to provide additional clarification if any questions remain unresolved.
> > >
> > > The workflows of existing diffusion-based unrestricted adversarial attack methods, except for the differences in the details of each method, are similar[1][2][3]. Since existing methods failed to generate unrestricted adversarial examples(UAEs) within the same framework under various scenarios, we are motivated to develop a unified framework with such a workflow and achieve UAEs generation under various attacking conditions by tuning some modules or hyperparameters of the framework.
> > >
> > > Although the flowchart in Figure 1 appears to have no significant difference after obtaining the low-level semantics, there are distinctions in the implementation of such two scenarios. For example, in the task of generating image-similar UAEs, we derive the posterior distribution under the adversarial constraint and reconstruction constraint, while in the task of generating UAEs from noise, since there’s no real image for reference, we derive the posterior distribution under the adversarial constraint and class condition y, hence the subsequent processes of the two scenarios like probability density function(PDF) inference and PDF maximization is also different. Besides, the diffusion models applied in such two scenarios are also different: the conditional latent diffusion model(LDM) for generating UAEs from noise and an unconditional diffusion model for generating image-similar UAEs.
> > >
> > > For the experiments proposed in AdvDiff, as is proposed in the last sentence of section 4.1 in their paper: ‘Note that we adopt the clean images generated by LDM to achieve DiffAttack and AutoAttack for a fair comparison’, which means the authors use noise images as inputs of the LDM to synthesize images, then use the synthesized images as clean image inputs of DiffAttack and AutoAttack to generate UAEs, such that the main contents of the UAEs generated by each method are the same. In other words, they employ the LDM before performing DiffAttack and AutoAttack, such that the input of their implementation of DiffAttack and AutoAttack is still noise, and the UAEs generated with each method maintain the same main contents. By leveraging such adaptation, methods compared within the same evaluation table are all noise-based. However, the attacking conditions of AdvDiff and DiffAttack are fundamentally different.
> > >
> > > In our method, the low-level semantics in generating UAEs from noise are randomly synthesized by the diffusion model. In generating image-similar UAEs, the low-level semantics are determined from real natural images. As a result, the content of UAEs generated under the two scenarios is completely different, such that we can not compare their naturalness and effectiveness in the same experiment. Besides, we reimplemented the source code of AdvDIff and DiffAttack without any adaptation, so we compare them in different tables under different scenarios. Moreover, AdvDiffuser also regards such two scenarios separately. From their paper, in section 4.2, they compare the UAEs generated from noise with AdvDiffuser and AC-GAN in Table 1, and in section 4.3, they compare image-similar UAEs with AdvDiffuser and GA-PGD in Tables 2 and 3.
> > >
> > > If our explanation has helped resolve your concerns, we would greatly appreciate it if you might consider revising your rating to acceptance.
> > >
> > > [1] AdvDiff: Generating Unrestricted Adversarial Examples using Diffusion Models.\
> > > [2] Advdiffuser: Natural adversarial example synthesis with diffusion models.\
> > > [3] Diffusion Models for Imperceptible and Transferable Adversarial Attack.

---

### Official Review · Reviewer_Y6Ps · 2025-03-13

**Overall Recommendation:** 3

**Summary:**

This paper proposes DiffAdvMAP, a Bayesian-based framework for generating universal adversarial examples (UAEs) by approximating their posterior distribution. Unlike existing diffusion-based methods, which struggle with low naturalness and limited effectiveness, DiffAdvMAP leverages adversarial and reconstruction constraints to enhance both attack success and adaptability.

**Claims And Evidence:**

Yes.

**Essential References Not Discussed:**

No.

**Ethical Review Concerns:**

No.

**Experimental Designs Or Analyses:**

(1) Multiple evaluation metrics are employed: FID and LPIPS for global image similarity UAEs, and FID, TRES, and HyperIQA for noise-based UAEs.
(2) Appropriate baselines are selected for different attack scenarios: AdvDiff for noise-based UAEs and classic unrestricted and diffusion-based attacks for global image similarity UAEs.
(3) Ablation studies analyze the impact of reconstruction constraints, adversarial constraints, destruction-reconstruction modules, and variations in adversarial confidence levels.

**Methods And Evaluation Criteria:**

Yes.

**Other Comments Or Suggestions:**

(1) Expand experiments by incorporating diverse datasets and industrial inspection images, to evaluate DiffAdvMAP’s adaptability. Investigate framework adjustments for specialized data and explore adversarial sample generation in multimodal scenarios.
(2) Optimize efficiency by improving diffusion model structures, refining sampling strategies, leveraging hardware acceleration, and exploring model compression and quantization to reduce resource consumption.

**Other Strengths And Weaknesses:**

(1) DiffAdvMAP is primarily evaluated on ImageNet-compatible datasets, but its adaptability to specialized domains like medical imaging and remote sensing remains unexplored.
(2) While the method improves generation speed, it still relies on complex diffusion models and multiple iterations, leading to high computational costs. This may hinder real-time attacks or large-scale data processing.

**Questions For Authors:**

(1) Expand experiments by incorporating diverse datasets and industrial inspection images, to evaluate DiffAdvMAP’s adaptability. Investigate framework adjustments for specialized data and explore adversarial sample generation in multimodal scenarios.
(2) Optimize efficiency by improving diffusion model structures, refining sampling strategies, leveraging hardware acceleration, and exploring model compression and quantization to reduce resource consumption.

**Relation To Broader Scientific Literature:**

Diffusion models excel in image synthesis and time series prediction, but their application in adversarial sample generation often relies on them as priors without fully utilizing their ability to learn real data distributions. DiffAdvMAP integrates adversarial and reconstruction constraints to better leverage learned priors, improving naturalness, attack success, and flexibility, expanding diffusion models’ role in adversarial attack research.

**Theoretical Claims:**

(1) In the Bayesian framework, given the reference image, true label, diffusion model, and target classifier, a mathematical formulation for UAE generation is constructed. The adversarial constraint is derived from the CW attack, and the reconstruction constraint is introduced based on different generation scenarios. The posterior distribution of UAEs is obtained using Bayesian principles, considering the properties of the diffusion model.
(2) For image-similar UAEs, the posterior distribution's PDF is derived, leading to the objective function through approximation. To balance computational complexity, a one-step estimation method is used, approximating the conditional distribution accordingly. The roles of key parameters, such as standard deviation adjustments, are clarified to ensure the correctness and practicality of the objective function.

---

> ### Author Rebuttal · Authors · 2025-04-01
>
> Thank you for the reviews and kindly suggestion. According to your opinions, we add experiments on another dataset Celeba-HQ, the experimental results are satisfactory. We wish you can improve the score after reading our response.\
> 1. Dataset Diversity: We use the ImageNet-compatible dataset to evaluate DiffAdvMAP because it contains the most important objects in the real world, and it is widely used in previous works[1][2] as a single dataset. We also evaluate DiffAdvMAP with another subset of ImageNet in Appendix D. DiffAdvMAP is developed based on pre-trained diffusion models, and since it’s hard to obtain open-sourced diffusion models in the area of medical imaging and remote sensing, we supplement experiments on the Celeba-HQ dataset in attacking the face gender classification model. The experimental details are as follows:\
> We trained a Resnet18 and a Mobilenet V2 as the gender classifier, and used the Resnet18 as the known surrogate classifier to generate 1000 UAEs, among them, 500 are female faces and 500 are male faces originally. For the evaluation, we propose the clean accuracy of each model, the white-box attack success rate(ASR) towards Resnet18, and transfer ASR towards Mobilenet V2 in the table below.\
> |Models   | Resnet18| Mobilenet V2|\
> |Clean Accuracy|98.33%|98.50%|\
> |ASR|100%|92.3%|\
> Note that the generation speed is still 6 seconds/image. We think this experiment can show the adaptability of DiffAdvMAP to other datasets to some extent.\
> [1] Dong Y, Pang T, Su H, et al. Evading defenses to transferable adversarial examples by translation-invariant attacks[C]//Proceedings of the IEEE/CVF conference on computer vision and pattern recognition. 2019: 4312-4321.\
> [2] Gao L, Zhang Q, Song J, et al. Patch-wise attack for fooling deep neural network[C]//Computer Vision–ECCV 2020: 16th European Conference, Glasgow, UK, August 23–28, 2020, Proceedings, Part XXVIII 16. Springer International Publishing, 2020: 307-322.
>
> 2. Optimize efficiency: As introduced in Section 1, diffusion models have a supreme performance in image synthesis. Our DiffAdvMAP method fully utilizes the prior knowledge of real data distribution learned by the diffusion models and manages to generate more effective and natural UAEs than the previous method. We already refined the sampling strategy by leveraging a truncated sampling process. We also propose targeted optimizations on our method to address the inherent generation speed constraints of other diffusion-based methods, keeping a comparable efficiency compared with most attack methods, which is shown in Table 2. Note that the GPU we applied to generate UAEs is RTX 3090 24G. The generation speed can be further improved if we use more advanced hardware. Use a lighter diffusion model to accelerate the generation speed is also included in our future research considerations.
>
> 3. Multimodal scenarios: In this paper, we are focusing on generating general UAEs in single-modal cases. Generating UAEs under the multimodal scenarios is an interesting area; we will conduct research in this area in our future work.

---

### Official Review · Reviewer_58n5 · 2025-03-17

**Overall Recommendation:** 3

**Summary:**

This paper introduces DiffAdvMAP, a flexible diffusion-based framework for generating Universal Adversarial Examples (UAEs) under various attack conditions. By approximating the posterior distribution of UAEs using pre-trained diffusion models, DiffAdvMAP generates more natural UAEs compared to existing diffusion-based attack methods. The framework incorporates an adversarial constraint to ensure attack effectiveness and a reconstruction constraint to enhance adaptability across different attack scenarios. Experimental results show that DiffAdvMAP outperforms baselines in white-box success rate, transferability, and defense robustness, demonstrating its superior flexibility and effectiveness.

**Claims And Evidence:**

Yes, the claims on the flexibility, naturalness and effectiveness of DiffAdvMAP to generate UAEs are supported by generally clear and convincing evidence.

The claim of the flexibility of DiffAdvMAP is supported by the design of this method in Constraint 2, i.e., C2 in Equation (2). It allows for different scenarios in UAE generation, including style generation and color generation. The qualitative evaluations in Experiments also indicate the flexibility can be empirically achieved.

The authors also claimed naturalness and effectiveness of DiffAdvMAP. Intuitively and mathematically, Equations (7), (8), (9) and (14) generally demonstrate the control on realism. Equation (4) controls the effectiveness of DiffAdvMAP to attack classifiers. Results in Tables 1 and 2 demonstrate the naturalness and effectiveness of generated UAEs empirically.

However, for the efficiency claimed for DiffAdvMAP, the experimental results in Tables 1 and 2 seem not to strongly support it. Compared to the past methods, time cost of DiffAdvMAP is not the SOTA.

**Essential References Not Discussed:**

For the defense methods against adversarial attacks, a recent and important paper: CLIPure [1] should be mentioned and discussed, because it exploits the semantic information in adversarial examples and achieve the SOTA performance, on top of DiffPure.

[1] MingKun Zhang, Keping Bi, Wei Chen, Jiafeng Guo, and Xueqi Cheng, CLIPure: Purification in Latent Space via CLIP for Adversarially Robust Zero-Shot Classification, ICLR 2025.

**Experimental Designs Or Analyses:**

As aforementioned, the experimental designs are in general sound. The dataset is ImageNet which is ubiquitous to investigate the performance of adversarial example generation. Success rates are necessary for the basic effectiveness of attackers. FID-score, TRES and HYPERIQA are also proper evaluation criteria, because computing LPIPS score needs reference images. Qualitative results are also demonstrated properly.

**Methods And Evaluation Criteria:**

The proposed DiffAdvMAP is consistent with and able to resolve the motivations via the maximum a posterior. In order to achieve the realism, it is natural to sample adversarial examples directly from an approximated posterior distribution and use the prior knowledge of real data distributions learned by diffusion models, instead of imposing strong prior as diffusion models.

In terms of the flexibility, the constraint in Equation (6) seems to be sound where $\Omega(\cdot)$ allows the change of styles, colors and other scenarios to amend $x$ and generate desired UAEs.

ImageNet is a commonly applied dataset to investigate the performance of adversarial example generation. Success rates for attacks can evaluate the basic effectiveness of attackers. FID-score, TRES and HYPERIQA are also proper evaluation criteria, because computing LPIPS score needs reference images and DiffAdvMAP should be studied also for the cases without reference images.

**Other Comments Or Suggestions:**

List of typos:
- Page 1 Introduction Line 35-37 "Such approaches do not require perturbing real images wi0000000000000000000000th restricted perturbations," -> "Such approaches do not require perturbing real images with restricted perturbations,"

**Other Strengths And Weaknesses:**

**Strengths**:
- *Originality*: This paper proposes an original approach to adversarial attack generation by integrating diffusion models with Bayesian inference. Unlike conventional adversarial attacks that rely on fixed perturbations, DiffAdvMAP samples adversarial examples from a learned posterior distribution, leading to more natural unrestricted adversarial examples (UAEs). Flexibility and realism of the generation is also well addressed.

- *Clarity*: The paper is well-organized and clearly written, making it accessible to readers familiar with adversarial attacks and diffusion models. The motivation, methodology, and experiments are explained in a logical flow.

**Weakness**:
- *Quality*: The paper does not provide a thorough analysis of the computational cost of DiffAdvMAP compared to traditional adversarial attack methods. The time cost of DiffAdvMAP is also not the smallest among the compared past methods in experiments. The biggest limitation of DiffAdvMAP is its decreased effectiveness on unseen classifiers in black-box attack scenarios, although DiffAdvMAP relies on the pretrained diffusion models. It may indicate that DiffAdvMAP overfits to the structure of known classifiers, making it less effective in real-world settings where models are often unknown.

**Questions For Authors:**

1. Would the authors provide more insights either theoretically or empirically on why DiffAdvMAP is more effective and efficient compared with past methods?

2. For stronger purifiers such as CLIPure, how effective will the proposed  DiffAdvMAP be to attack them?

**Relation To Broader Scientific Literature:**

This work enhances adversarial attackers generation to generate more natural and flexible UAEs. It extends diffusion-based attacks beyond standard settings by allowing UAEs to be generated under various attack conditions (e.g., from noise, region-based modifications, color changes). DiffAdvMAP moves from latent-space perturbation (Chen et al., 2023) to posterior-based sampling, leading to more realistic adversarial images. It is also a more flexible adversarial attack framework than previous diffusion-based approaches.

**Theoretical Claims:**

This work does not have theoretical claims.

---

> ### Author Rebuttal · Authors · 2025-04-01
>
> Thank you for the reviews and kindly reminder. We added the theoretical explanation and the experiments on CLIPure as follows. We wish you can improve the score after reading the response.
> 1. Efficiency: The main motivation of the paper is to develop a framework that achieves superior naturalness and adversarial effectiveness while maintaining practical generation efficiency. Due to the inherent generation speed constraints of diffusion-based methods, the efficiency of DiffAdvMAP is not the best among the baseline methods. Still, DiffAdvMAP surpasses most of the baselines in attack efficiency, achieves optimal naturalness, and matches ReColorAdv in generation speed. It achieves the best trade-off between naturalness, efficiency, and effectiveness. \
> The theoretical analysis of efficiency:\
> (1) Generating image-similar UAEs: As introduced in Section 2, compared to other diffusion-based methods(DiffPGD, AdvDiffuser, DiffAttack): Standard diffusion models require multiple U-Net queries for each image generation, causing inherent slowness. DiffAdvMAP accelerates this by generating UAEs via a truncated generation process(20 U-Net queries), achieving 30% faster speeds than DiffPGD (30 queries) and 10× acceleration over AdvDiffuser(400 queries). Even with equivalent query counts to DiffAttack, our method's 1-2 MAP optimizations per step enable 4×faster processing than DiffAttack's attention map computations and optimizations. Compared to non-diffusion-based methods: While GANs have fewer parameters than U-Nets, cAdv requires hundreds of network queries, making it slower than some diffusion approaches. Iterative methods like NCF and ReColorAdv bypass network queries entirely—their speed depends on the objective function complexity and iteration count. \
> (2) Generating UAEs from noise: For the task of generating UAEs from noise, compared to AdvDiff, both of the methods utilize the same generation process of the latent diffusion model to generate UAEs. Since there’s no real image for reference, we employ a strong class guidance to ensure the naturalness of the UAEs, such that DiffAdvMAP should perform more MAP optimization steps to counteract the strong class guidance in generating adversarial features to ensure enough transferability of UAEs. From Table 1, we can see that DiffAdvMAP has better transferability and naturalness than AdvDiff, requiring only 10% more time.
> 2. Effectiveness: As illustrated in Sec.3, we introduce an adversarial constraint to ensure the effectiveness and a reconstruction constraint to control the content of generated UAEs. Such two constraints are considered as two events of the posterior distribution of UAEs. Then the posterior distribution is approximated with the prior knowledge of real data learned by the diffusion model. Unlike previous methods which only use diffusion models as strong denoisers to enhance the generation process, DiffAdvMAP generates adversarial features by sampling from the approximated posterior distribution, which make the adversarial features more coherent to the original features, thus improving the naturalness of UAEs. As a result, DiffAdvMAP have more space to improve effectiveness at the cost of sacrificing naturalness. As illustrated in Tab.6 in Appendix E, the white-box attack success rate is nearly 100%, but the transfer ASR is not enough when the absolute value of the confidence level is small. When the absolute value of the confidence level increases, the transferability increases, and although the naturalness decreases, it is still better than other baselines.\
> (1) We don’t think DiffAdvMAP overfits to the structure of known classifiers. \
> When performing a transfer-based black-box attack, UAEs are generated with a known surrogate model, eg. Inception V3, then these UAEs are used to attack other unknown classifiers eg. Mobilenet V2, Resnet50 and Swin-Transformer and obtain the transfer attack success rate, the principal of transfer attack results in the decrease of attack success rate(ASR). As shown in Table 2, all of the existing black-box attack methods face an ASR decrease when attacking unknown classifiers. \
> (2) We complement the experiment on a stronger purifier, CLIPure, with DiffAdvMAP and DiffPGD. The ASRs are as follows:\
> |Methods/Surrogate Models| Mobilenet V2| Swin-transformer| Incepetion V3 |Resnet50|\
> |DiffAdvMAP|  48.3  | 57.6  | 39.9  | 49.2|\
> |DiffPGD      |   31.7 |  52.4|  24.6   |33.8|
> CLIPure is a zero-shot classifier that classifies images by matching an image with text prompts. It also leverages the CLIP model to purify the adversarial examples in the latent space. The ASR demonstrates that DiffAdvMAP can change the semantic information of images coherently, such that it can resist the semantic purification in the latent space and mislead the text-to-image semantic-based classifier to make wrong predictions in a large proportion.
> 3. Typos: Very thankful for your kindly reminder. We will correct it in revised version.

---

> > ### Comment · Reviewer_58n5 · 2025-04-07
> >
> > I thank the authors for their explanation and added experimental results. All my questions and concerns are generally addressed. I will keep my score.

---

### Decision · Program_Chairs · 2025-05-01

**Decision:**

Accept (poster)

**Comment:**

This paper introduces a Bayesian framework to generate unrestricted adversarial examples (UAEs) by sampling from posterior distributions using a diffusion model as a prior. The method integrates adversarial and reconstruction constraints to balance attack effectiveness, image naturalness, and flexibility across diverse attack scenarios (e.g., noise-based or style-transfer). Experiments on ImageNet show superior performance compared to existing methods like AdvDiff. The work received comments from four reviewers. Three reviewers tend to accept the work due to the novel Bayesian framework with Diffusion models for UAE and a comprehensive evaluation. The positive reviewers are also concerned about the computational cost and the lack of theoretical analysis. After the rebuttal, the main concerns are addressed. One reviewer rates the rejection at the first round due to the overstated novelty and omitted baselines. After the rebuttal, the reviewer acknowledges the changes and raises the score to weak reject. The remaining concern is the necessity of separating the generation of UAEs from noise-based and image-similar cases. Upon my own review of the paper, I understand this concern does not significantly detract from the work's originality. Using a single method to address both scenarios offers the practical advantage of reducing the overhead that would come with implementing two independent models. Due to the above facts, I recommend accepting this work.